

# The impact of Aeolus winds on surface wind forecast over tropical ocean and high latitude regions

Haichen Zuo[1], Charlotte Bay Hasager[1]

[1]Wind and Energy Systems, Technical University of Denmark, Roskilde, 4000, Denmark

*Correspondence to*: Haichen Zuo (hazu@dtu.dk)

**Abstract.** To detect global wind profiles and improve numerical weather prediction (NWP), the European Space Agency (ESA) launched the Aeolus satellite carrying a space-borne Doppler Wind Lidar in 2018. After the successful launch, the European Centre for Medium-Range Weather Forecasts (ECMWF) performed the observing system experiments (OSEs) to evaluate the contribution of Aeolus data to NWP. This study aims to assess the impact of Aeolus wind assimilation in the

ECMWF model on surface wind forecast over tropical ocean regions by taking buoy measurements for reference and over high latitude regions by taking weather station data for reference for the year 2020. The assessments were conducted through inter-comparison analysis and triple collocation analysis. The results show that with Aeolus data assimilation, the tropical sea surface wind forecast could be slightly improved at some forecast time steps. The random errors of u (zonal) and v (meridional) wind components from OSEs are within 1 m s$^{-1}$ with respect to the model resolution. For the high latitude

regions, Aeolus can reduce the wind forecast errors in the Northern Hemisphere with forecast extending, particularly during the first half-year of 2020 and during the winter months. For the Southern Hemisphere, the positive impact is mainly found for the u component at most forecast steps during June, July and August. Moreover, compared with the tropical ocean regions and the region > 60° N, the random error of OSEs for the region > 60° S increases significantly to 3 m s$^{-1}$ with forecast extending. Overall, this study demonstrates the ability of Aeolus winds to improve surface wind forecast over

tropical oceans and high latitude regions, which provides valuable information for practical applications with Aeolus data in the future.

## 1 Introduction

For characterizing global wind profiles and improving numerical weather prediction (NWP), the first space-borne Doppler Wind Lidar (DWL) carried by the Aeolus satellite was launched in August 2018 by the European Space Agency (ESA). The

mission is still ongoing and has been operating for more than four years. Following a sun-synchronous orbit, Aeolus passes over the equator at 06:00 local time (LT) during descending orbits and 18:00 LT during ascending orbits and samples the whole globe every twelve hours with eight orbits. Wind retrieval of Aeolus is based on the Doppler shifted frequency between emitted light pulses and backscattered light from air molecules (i.e. Rayleigh scattering) as well as from large particles, such as cloud droplets and ice crystals, in the atmosphere (i.e. Mie scattering). By measuring this small difference,





wind velocity along the line-of-sight (LOS) can be obtained, which is further converted to the approximately east-west horizontal LOS wind component using the off-nadir angle of 35° (Andersson et al., 2008). The detected wind profiles, ranging from the surface to about 30 km in height with 24 vertical bins, can be used to improve NWP, capture gravity waves, track volcanic eruptions, etc. (Banyard et al., 2021; Rennie et al., 2021; Parrington et al., 2022).

After the successful launch, calibration and validation works have been widely carried out worldwide. Owing to the continually improved data processing chain, from Baseline 10 with M1-temperature-based bias correction and daily updates of global offset bias removal (Data Innovation and Science Cluster, 2020), the systematic errors of both Rayleigh-clear winds and Mie-cloudy winds are almost within 0.5 m s$^{-1}$ despite some cases in the polar regions, and the random errors mainly vary between 4 m s$^{-1}$ and 8 m s$^{-1}$ for Rayleigh-clear winds and between 2.0 m s$^{-1}$ and 5 m s$^{-1}$ for Mie-cloudy winds
(Belova et al., 2021; Iwai et al., 2021; Witschas et al., 2022; Zuo et al., 2022). However, what should be noted is that Aeolus has been suffering unexpected signal loss since the launch, probably due to the decreasing emitted laser energy for the FM-A period (August 2018 – June 2019) and/or laser-induced contamination for the FM-B period (July 2019 – September 2022) (Straume-Lindner et al., 2021). The data quality assessment based on the second reprocessed data set (2B11) by the European Centre for Medium-Range Weather Forecasts (ECMWF) revealed that the estimated random error of Rayleigh-
clear wind increased by 40% from ~5 m s$^{-1}$ to ~7 m s$^{-1}$ during July 2019 – October 2020 due to the gradual signal reduction of DWL, while this instrument issue has less influence on Mie-cloudy winds with estimated random errors remaining at ~3.5 m s$^{-1}$ (Rennie and Isaksen, 2022).

Although Aeolus suffers from unexpected signal loss and growing errors, its wind products have been employed to improve
NWP through data assimilation, an approach that integrates recent observations with a previous forecast to achieve the best estimate of the current atmospheric state (ECMWF, 2020). For evaluating the contribution of Aeolus observations to NWP, the observing system experiments (OSEs) with and without Aeolus data assimilation have been performed with global NWP models at many institutions, including the ECMWF, National Oceanic and Atmospheric Administration (NOAA), Deutscher Wetterdienst (DWD), Météo-France, UK Met Office, etc. (Cress et al., 2022; Garrett et al., 2022; Forsythe and Halloran,
2022; Pourret et al., 2022; Rennie and Isaksen, 2022). The OSEs with the ECMWF model demonstrated that Aeolus winds are able to improve wind vector and temperature forecasts, especially in the upper troposphere and/or lower stratosphere over tropical and polar regions (Rennie et al., 2021). Similar results were also found from the OSEs with NOAA's Global Forecast System, the DWD model and the Environment and Climate Change Canada global forecast system (Cress et al., 2022; Garrett et al., 2022; Laroche and St-James, 2022). Moreover, regarding the weather and climate events, Aeolus is able
to improve the track forecasts for tropical cyclones in the Eastern Pacific basin and Atlantic basin (Garrett et al., 2022). In addition to this, Aeolus benefits the forecasts of the West African Monsoon and the changes in the El Niño-Southern Oscillation (ENSO) state over the Eastern Pacific by capturing the large-scale atmospheric circulation (Cress et al., 2022). However, the existing assessments mainly focused on the forecasts from pressure levels, while the impacts of Aeolus data



assimilation on surface wind forecast over tropical ocean regions and polar regions lack detailed study. This research gap

needs to be complemented since the relevant scientific investigation could provide valuable information for wind-related

activities, such as ocean shipping and wind farm operation and maintenance.

Regarding the reference dataset for evaluation, many verifications related to Aeolus OSEs were conducted by inter-comparing with model analysis (Garrett et al., 2022; Pourret et al., 2022; Rennie and Isaksen, 2022). Since there are fewer in

situ measurements available over tropical and polar regions, the analysis data may have large uncertainties in these regions. In terms of the evaluation method, apart from the conventional inter-comparison analysis, triple collocation analysis is another beneficial method for environmental parameter evaluation when there are three independent data sets (Stoffelen, 1998; Vogelzang and Stoffelen, 2012). Different from the inter-comparison analysis that regards a reference data set free of errors, triple collocation (TC) analysis assumes that each system is linearly correlated with the truth. Furthermore, TC

analysis takes representative errors into account, which is advantageous when the reference data set has different spatial and temporal resolutions with the data sets to be assessed. The primary outputs of TC include the error standard deviation (or random error) of each system and calibration coefficients based on a reference data set (Vogelzang and Stoffelen, 2012). TC method has been widely implemented to characterize the errors for wind measurements from scatterometers, altimeters, and DWL, etc. (Caires and Sterl, 2003; Vogelzang et al., 2011; Ribal and Young, 2020; Cossu et al., 2022). However, to the

author's best knowledge, no studies have evaluated wind forecasts by this method so far.

Hence, to complement the existing studies, this study aims to assess the impact of Aeolus wind assimilation on surface wind forecast over tropical ocean regions between 30° N and 30° S by taking buoy measurements for reference. Furthermore, we investigate the high latitude region > 60 °N in the Northern Hemisphere (NH) and the high latitude region > 60° S in the

Southern Hemisphere (SH) by taking weather station data for reference. Our hypothesis is that assimilation of Aeolus winds will reduce the forecast error. Since the overall data quality of Aeolus is reduced in the second half-year of 2020 compared to the first half-year due to the weakening signals, our hypothesis is that the assimilation of Aeolus winds can reduce the forecast error relatively more during the first half year compared to the second half-year. In the tropics, seasonal effects are very limited, while in the high latitude regions, the seasonal variability is high, so for those we also investigate the forecast

for the seasons, mainly for the summer and winter months of each hemisphere. The assessments were conducted through both inter-comparison analysis and triple collocation analysis based on a high resolution T639 OSE in the ECMWF model for the entire year of 2020.

Section 2 and Sect. 3 introduce the data and methods used in this study. Sect. 4 presents the main research findings, followed

by Sect. 5 for discussions. The final Sect. makes a short summary of the study and draws conclusions.



## 2 Data

### 2.1 Observing System Experiments with ECMWF model

This study is based on the OSEs with the ECMWF model by using the 2nd reprocessed Aeolus L2B baseline 11 data during the FM-B period (Rennie and Isaksen, 2022). The applied model version is CY47R2 with an atmosphere resolution of $T_{co}$

639 L137 (~ 18 km grid). Observations from nominally operational satellites and conventional sources were assimilated. The OSEs include a control experiment without Aeolus assimilation and an experiment with Aeolus Rayleigh-clear and Mie-cloudy wind assimilation through the four dimensional variational (4D-Var) data assimilation technique. The 10-day forecasts of zonal (u) and meridional (v) wind components at 10 m height were extracted from the Meteorological Archival and Retrieval System (MARS) for evaluation (ECMWF Research Department, 2022). The interval of forecast steps is 24

hours. The data cover the completed year of 2020.

### 2.2 Buoy measurements

The tropical moored buoy measurements over the Atlantic Ocean, Indian Ocean and Pacific Ocean were obtained from Global Tropical Moored Buoy Array (Pacific Marine Environmental Laboratory, n.d.). The extracted parameters include zonal (u) and meridional (v) wind components, wind speed, and wind direction with a temporal resolution of 10 minutes or 1

hour. The missing value and data flagged low-quality have been removed. Finally, there are 11 buoys available in the Atlantic Ocean, 9 in the Indian Ocean and 55 in the Pacific Ocean, the locations of which are displayed in Fig. 1. To make all measurements have an identical temporal resolution, we averaged the 10 minutes wind speeds to hourly wind speeds. Furthermore, to collocate with wind forecasts from OSEs, the buoy winds were extrapolated from its anemometer height of 3.5 m or 4 m to 10 m by using the method described in Bidlot et al. (2002).

### 2.3 Weather station data

Surface synoptic observations over high latitude regions (> 60° N and > 60° S) were extracted from Global Hourly - Integrated Surface Database (ISD) (National Centers for Environmental Information, n.d.). Only the wind speeds and directions passed quality control checks were kept for further analysis. Additionally, we calculated the correlation coefficients (R) between in situ measurements and the control experiments at T+0 h, and the stations with weak correlations

(R < 0.5) were removed. After quality control, there are 751 and 56 stations available over the high latitude regions in the Northern and Southern Hemisphere, respectively (Fig. 1).



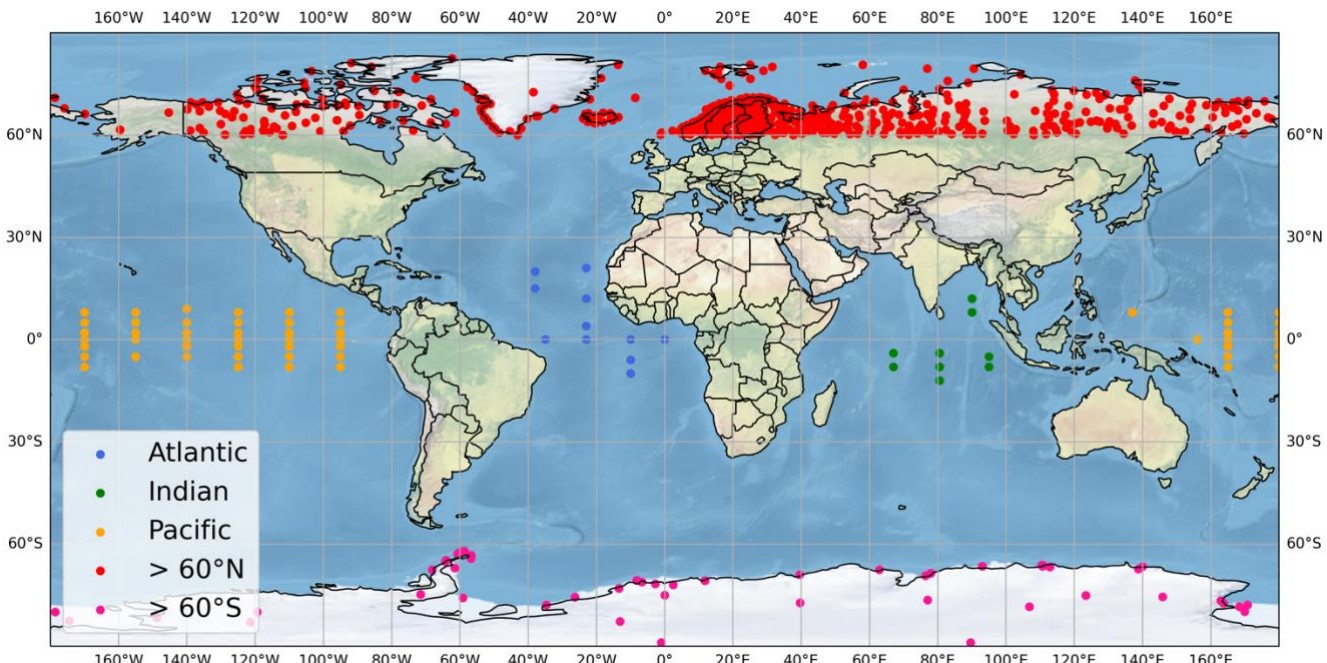

**Figure 1: The geographical location of moored buoys in the tropic oceans and weather stations in the high latitude > 60° N and high latitude > 60° S (background image made with Natural Earth. Free vector and raster map data at naturalearthdata.com).**

## 3 Method

### 3.1 Inter-comparison analysis

To evaluate the wind forecasts from OSEs, we take buoy measurements or weather station observations for reference, respectively. We quantified the normalized difference between the root mean square errors (RMSE) with and without Aeolus data assimilation for all paired data samples, thus determining whether the Aeolus can improve the model performance or not over each study region. The normalized difference between RMSEs (NDRMSE) is given as

$$NDRMSE = \frac{\sqrt{\frac{\sum_1^N (f_{with\ Aeolus} - o_{In\ situ})^2}{N}} - \sqrt{\frac{\sum_1^N (f_{no\ Aeolus} - o_{In\ situ})^2}{N}}}{\sqrt{\frac{\sum_1^N (f_{no\ Aeolus} - o_{In\ situ})^2}{N}}} \tag{1}$$

where $f_{with\ Aeolus/no\ Aeolus}$ is the wind forecast with/without Aeolus data assimilation; $o_{In\ situ}$ is the in situ measurements from either buoys or weather stations; and N is the total number of paired data samples for each study region or each case.

### 3.2 Triple collocation analysis

In addition to inter-comparison analysis, we conducted the evaluation via triple collocation (TC) analysis by taking the in situ measurements as the truth, defined as system 1 in this study. Systems 2 and 3 are the control experiment without Aeolus





data assimilation and the experiment with Aeolus data assimilation, respectively. Table 1 gives the temporal and spatial resolutions of these three systems. Eq. (2), (3) and (4) describe the linear relation of each system with the true wind.

$$U_1 = t + e_1 \tag{2}$$

$$U_2 = a_2 + b_2 t + e_2 \tag{3}$$

$$U_3 = a_3 + b_3 t + e_3 \tag{4}$$

$U_i$ is the wind component (u or v) of each system; $t$ is the true value of a wind component; $a_i$ and $b_i$ are the intercept and the gradient of the calibration for each system; $e_i$ is the error standard deviation or random error of each system.

**Table 1. Spatial and temporal resolution of the three systems**

|  | **1: In situ** | **2: Experiment (No Aeolus)** | **3: Experiment (With Aeolus)** |
|---|---|---|---|
| Horizontal | Point-based | ~ 18 km | ~ 18 km |
| Vertical | 10 m | 10 m | 10 m |
| Temporal | hourly | Instantaneous | Instantaneous |

Detailed information on the TC method and its related equation derivation can be found in Stoffelen (1998) and Vogelzang and Stoffelen (2012). According to the resolution information in Table 1, there is no common signal of System 1 (in situ measurements) and System 2 or 3 (control experiment or experiment with Aeolus data) not resolved by System 3 or 2 (experiment with Aeolus data or control experiment), which implies that the error covariances are free of representation error. The simplified equations for quantifying the error standard deviation are given in Eq. (5), (6) and (7).

$$\sigma_1 = \sqrt{\langle e_1^2 \rangle} = \sqrt{C_{11} - \frac{C_{12}\,C_{13}}{C_{23}}} \tag{5}$$

$$\sigma_2 = \sqrt{\langle e_2^2 \rangle} = \sqrt{C_{22} - \frac{C_{12}C_{23}}{C_{13}}} \tag{6}$$

$$\sigma_3 = \sqrt{\langle e_3^2 \rangle} = \sqrt{C_{33} - \frac{C_{23}C_{13}}{C_{12}}} \tag{7}$$

where $C_{ii}$ is the variance of each system, and $C_{ij}$ is the covariance between the system i and j; and $\langle\ \rangle$ represents the arithmetic mean; $\frac{C_{12}\,C_{13}}{C_{23}}$ is the common true variance, denoted as CTV in the plots hereafter.

The analyses were performed for each ocean basin, regions $> 60°$ N and $> 60°$ S, respectively. We also divided the study period into two half-years to evaluate the sensitivity of Aeolus data quality on wind forecast. For high-latitude regions, the study was also carried out for each season. Since the error model of TC works better for noisy u and v wind components than wind speed (Stoffelen, 1998), the triple collocation analyses were only performed for u and v wind components. In addition, TC analyses were carried out based on all paired data samples over each study region rather than from each site since the optimal number of data samples for TC is at least 1000. To be consistent with TC, inter-comparison analyses were also





performed based on all paired data samples. The statistical significance of NDRMSE was quantified at the 95% confidence interval (not shown on plots).

## 4 Results

### 4.1 Tropical oceans

### 4.1.1 Inter-comparison analysis

The inter-comparison analysis shows that with Aeolus wind assimilated, the forecasts on u, v vectors and wind speed can be slightly and randomly improved over all three ocean basins. Note, improvement in the forecast is given in negative values as the error is reduced in the forecasting with Aeolus assimilated. For the tropical Atlantic Ocean, the improvement on the u

vector is mainly at T+48 h, 120 h and 240 h, while the improvement on the v vector concentrates within 72 h; the NDRMSEs of wind speed forecast are reduced at the first 72 hrs, 122 h and 144 h and 240 h. For the Indian Ocean, the forecast improvement is mainly at T+48 h, 96 h, 120 h and even 240 h for the u vector and at T+168 h for the v vector; positive impacts on wind speed forecast are mainly found at T+48 h, 96 h, 120 h and even 240 h. Compared with the tropical Atlantic and the Indian Ocean, the Pacific witnesses the forecast improvement at more forecast steps, but the magnitude is weaker.

Unfortunately, the NDRMSEs are not statistically significant at a 95% confidence interval for all three tropical ocean regions.





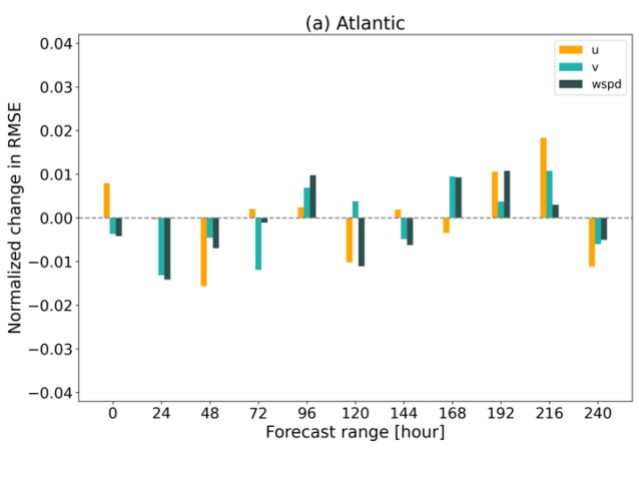

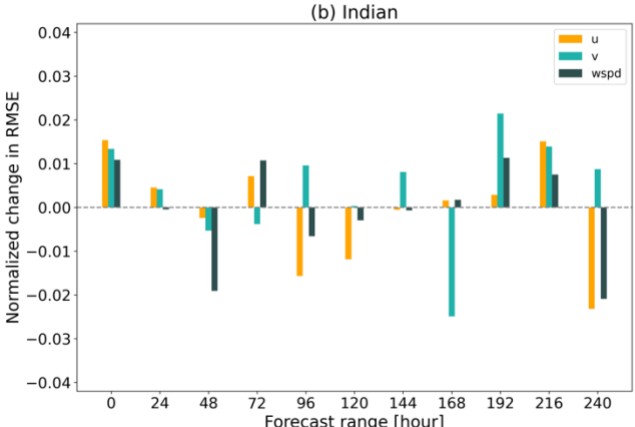

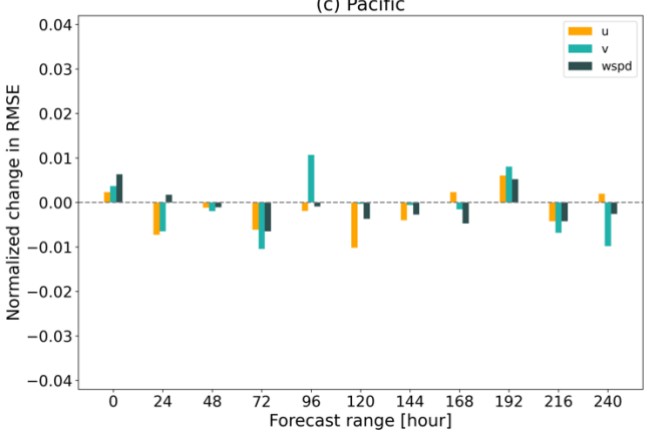


**Figure 2: Normalized difference in RMSE for u, v wind components and wind speed (wspd) for the tropical Atlantic (a), Indian Ocean (b) and Pacific (c) for the year 2020 based on ECMWF OSE forecasts with and without Aeolus against buoy data. Note that negative values indicate error reduction, implying the improvement in the forecast with Aeolus assimilation.**





Regarding the results for two-half years during the study period (Fig. 3), the impacts of Aeolus data quality during different
periods on u, v wind components and wind speed forecast are limited over tropical ocean regions. Only for the v component
over the tropical Atlantic, there are more error reductions from T+48 h to T+168 h during the first half-year; and over the
Indian Ocean, the first half-year from T+96 h to T+192 h witnesses more error reduction for the forecast on u component.

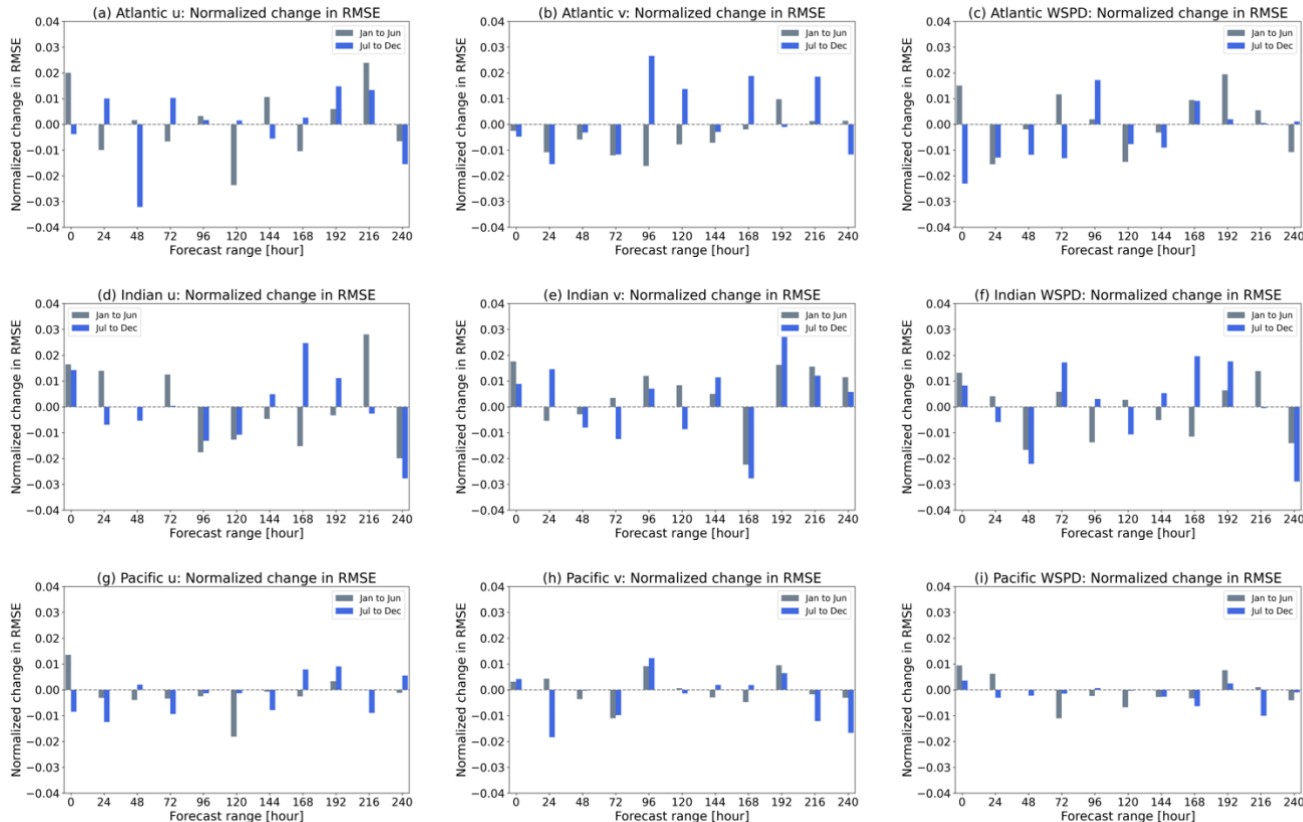


**Figure 3: Normalized difference in RMSE for u, v wind components and wind speed (wspd) during the first and the second half-year of 2020 for the tropical Atlantic (a, b and c), Indian Ocean (d, e and f) and Pacific (g, h and i) based on the ECMWF OSE forecasts with and without Aeolus data assimilation.**

### 4.1.2 Triple collocation analysis

The results from the triple collocation analysis are shown in Figure 4. With the forecast time extending, the errors from the
OSEs and buoys increase slightly for both u and v components, but the errors (usually $< 1$ m s$^{-1}$) from the two experiments
are much lower than the buoy errors ($> 1$ m s$^{-1}$) for all three ocean basins. The common true variance decreases with forecast
time for all cases, and large values (above 18 m$^2$ s$^{-2}$) are for the v component for the Atlantic and for the u component for the
Indian Ocean. Based on the results in Fig. 4, the impacts of Aeolus data assimilation on the forecasts for the tropical ocean
basins are nearly neutral. In terms of the results for two half-year periods (Fig.5), there are slight error reductions of u and v
wind components for the Atlantic region, mainly from T+72 h to T+168 h during the first half-year of 2020. For the Indian
and Pacific Oceans, the impacts of Aeolus data quality seem more limited (Appendix A).







**Figure 4: Error standard deviation and common true variance (CTV) of u, v wind components from triple collocation for the tropical Atlantic (a and b), Indian Ocean (c and d) and Pacific (e and f) for the year of 2020 based on the ECMWF OSE forecasts with and without Aeolus data assimilation and buoy data.**



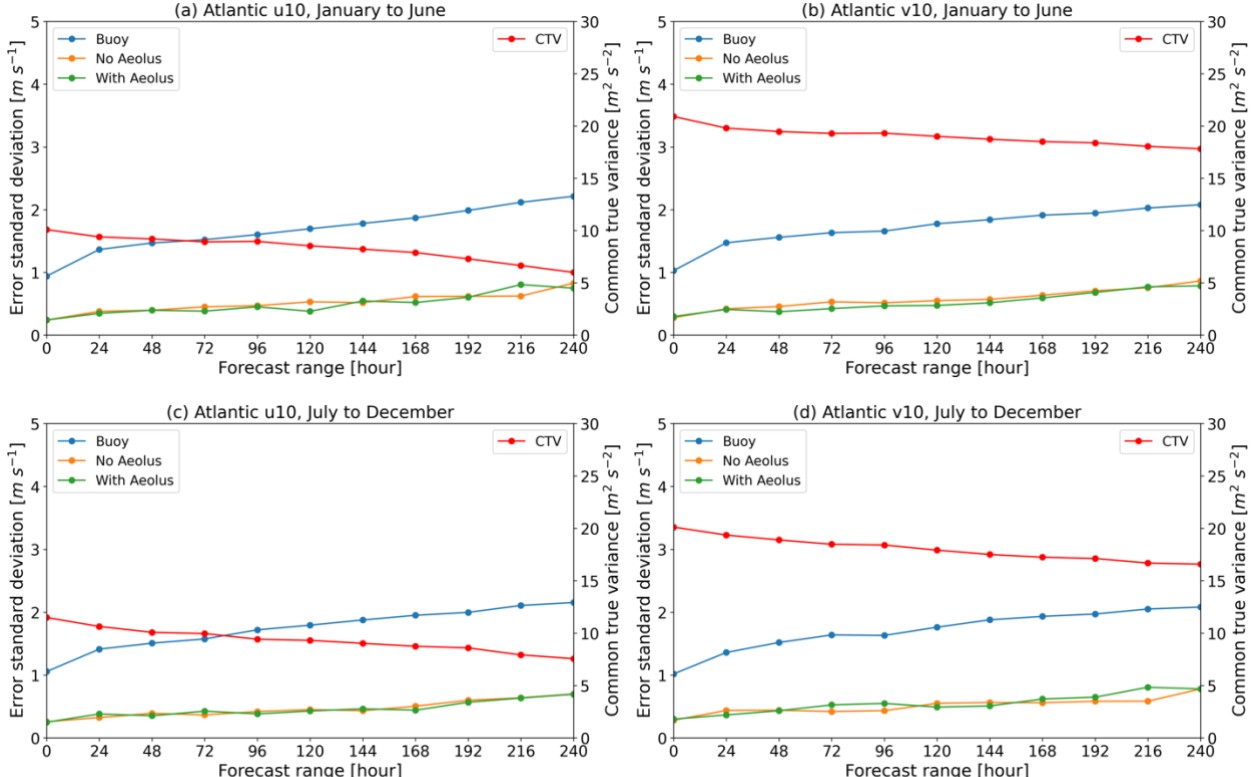

**Figure 5**:**Same as Fig. 4 but for the first (a and b) and the second half-year (c and d) of 2020 over the tropical Atlantic.**

### 4.1.3 Correlation

The correlation coefficients were calculated between each two systems to facilitate result interpretation. The results show that the forecast experiment with and without Aeolus is highly correlated up to T+120 h, with R-values greater than 0.9 for both u and v components as well as wind speed. As the forecast extends, the correlations between each two systems weaken but do not decrease too much for tropical ocean basins with R values greater than 0.7 at T+240 h for most cases. Figure 6 is an example for the tropical Pacific at T+120 h forecast step. The results show the u and v components with R-values around 0.95 for the forecasts with and without Aeolus, while for wind speed, R is around 0.90. R-values for u and v versus buoy data are around 0.8 and around 0.64 for wind speed for both forecasts. In summary, the zonal and meridional wind components are better resolved in the forecast model than the wind speed. The correlations do not reveal much improvement in forecast skill between the two forecasts. Similar results are found for the Atlantic and Indian Oceans (not shown).





**Figure 6: Hexagonal binning plots of u, v components and wind speed (wspd) at T+120 hour forecast for the tropical Pacific for the year 2020 based on ECMWF OSE forecasts with and without Aeolus and buoy data. The colour of each hexagon indicates the number of samples in it.**

## 4.2 High latitude region in the Northern Hemisphere (> 60° N)

### 4.2.1 Inter-comparison analysis

Over the high latitude region in Northern Hemisphere, the normalized differences in RMSE for u, v components and wind speed are almost negative and decrease as the forecast time extends, which implies Aeolus has a positive impact on surface wind forecast (Fig. 7). In particular, the changes for v vector were greater than 1% from T+192 h. Moreover, the forecast



errors of u, v component reduce more since T+120 h during the first half-year than that of the second half-year (Fig.8 (a) and (b)). This indicates that Aeolus's data quality is important for these forecasts. With respect to the normalized difference in RMSE for reach season (Fig.9), Aeolus provides a larger impact from T+120 h onwards to the u component forecasts during boreal winter (January, February and December) than during the boreal summer (June, July and August). For the v

component, the most noticeable error reductions (> 3.5%) exist at T+168 h during winter months and T+216 h during spring (March, April and May). Apart from these, no significant differences caused by seasons were observed.

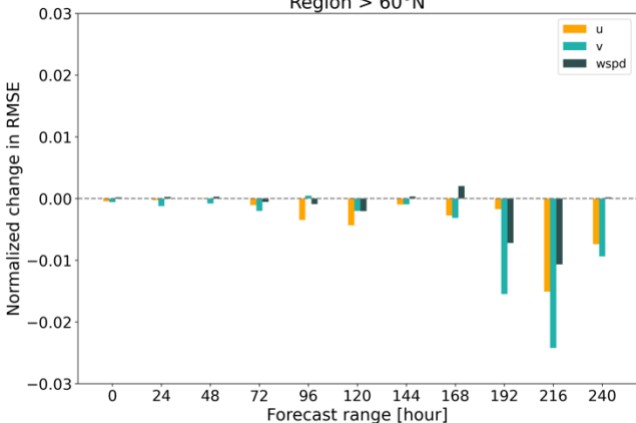

**Figure 7: Normalized difference in RMSE of u, v components and wind speed (wspd) as a function of forecast range for the region > 60° N for the year 2020 based on the ECMWF OSE forecasts with and without Aeolus and weather station data.**





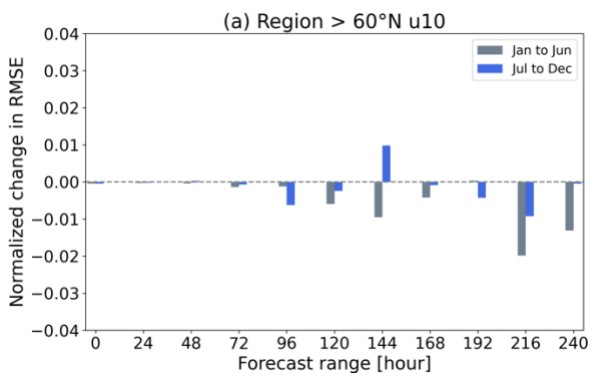


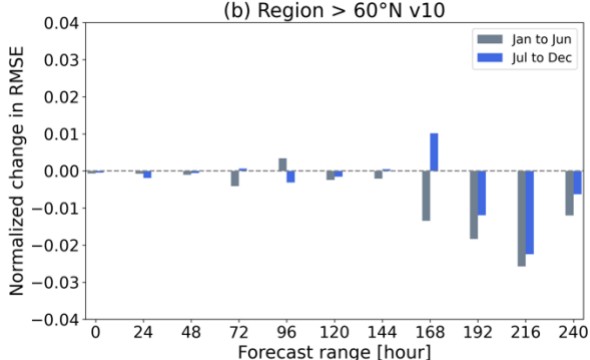

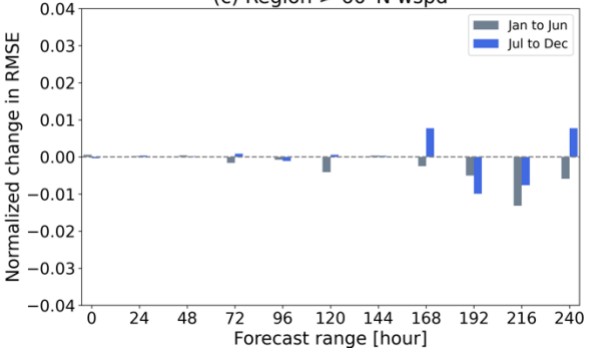

**Figure 8: Normalized difference in RMSE of u, v components and wind speed (wspd) as a function of forecast range during each half-year of 2020 for the region > 60° N based on the ECMWF OSE forecasts with and without Aeolus and weather station data.**






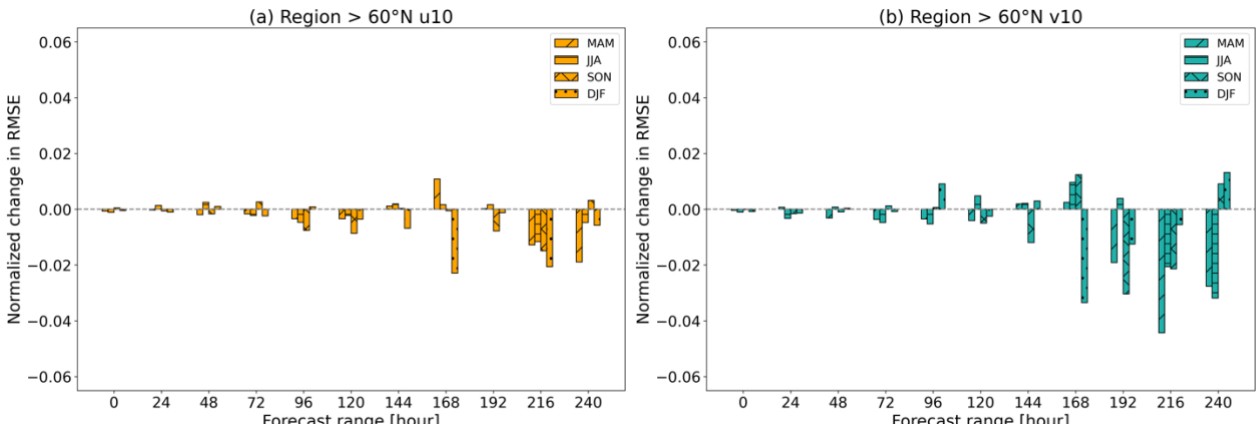

**Figure 9: Seasonal normalized difference in RMSE of u, v components as a function of forecast range for the region > 60° N for the year 2020 based on the ECMWF OSE forecasts with and without Aeolus and weather station data. MAM: March, April, and May; JJA: June, July, and August; SON: September, October, and November; DJF: December, January, and February.**

### 4.2.2 Triple collocation analysis

The results of triple collocation analysis for the region > 60° N are given in Fig. 10, 11 and 12. With forecast extending, the error standard deviations of u and v components from OSEs increase gradually from ~ 0.2 m s$^{-1}$ to ~ 1 m s$^{-1}$, but the values from the experiments with Aeolus are slightly smaller than the ones without Aeolus assimilation, particularly from T+168 h for u component and T+192 h for v component. The error standard deviations of in situ measurements also increase from ~1.8 m s$^{-1}$ to ~ 3 m s$^{-1}$ for both u and v components, while the common true variances decrease to below 2 m$^2$ s$^{-2}$. With respect to the results for each half-year (Fig. 11), there are more error reductions for both u and v components during the first half-year with Aeolus data assimilation, and the forecast improvement exists earlier from T+144 h for the u component and T+168 h for the v component. In terms of the seasonal variation (Fig. 12), Aeolus tends to have more positive impacts on the mid-range forecast during boreal winter with smaller errors from T+144 h for the u component and from T+168 h for the v component.

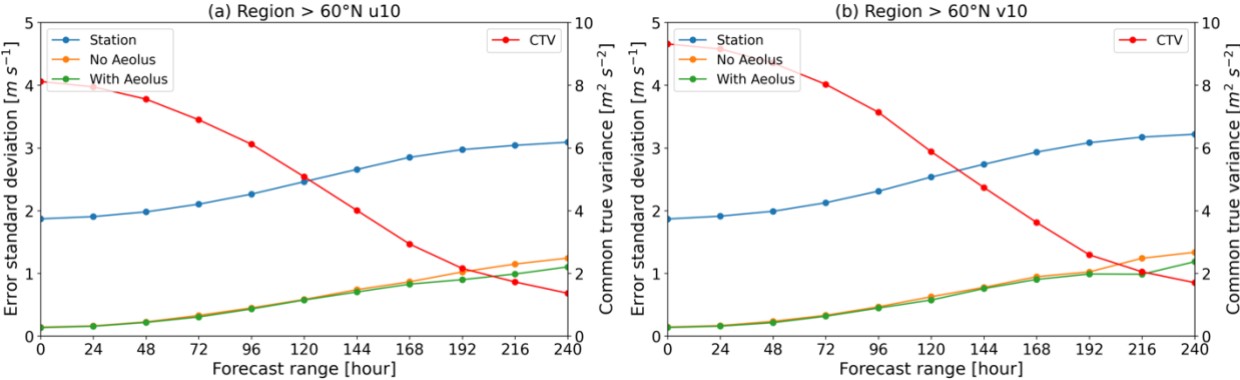

**Figure 10: Error standard deviation and common true variance (CTV) of u, v components and wind speed (wspd) as a function of forecast range for the region > 60° N for the year 2020 based on the ECMWF OSE forecast with and without Aeolus and weather station data.**




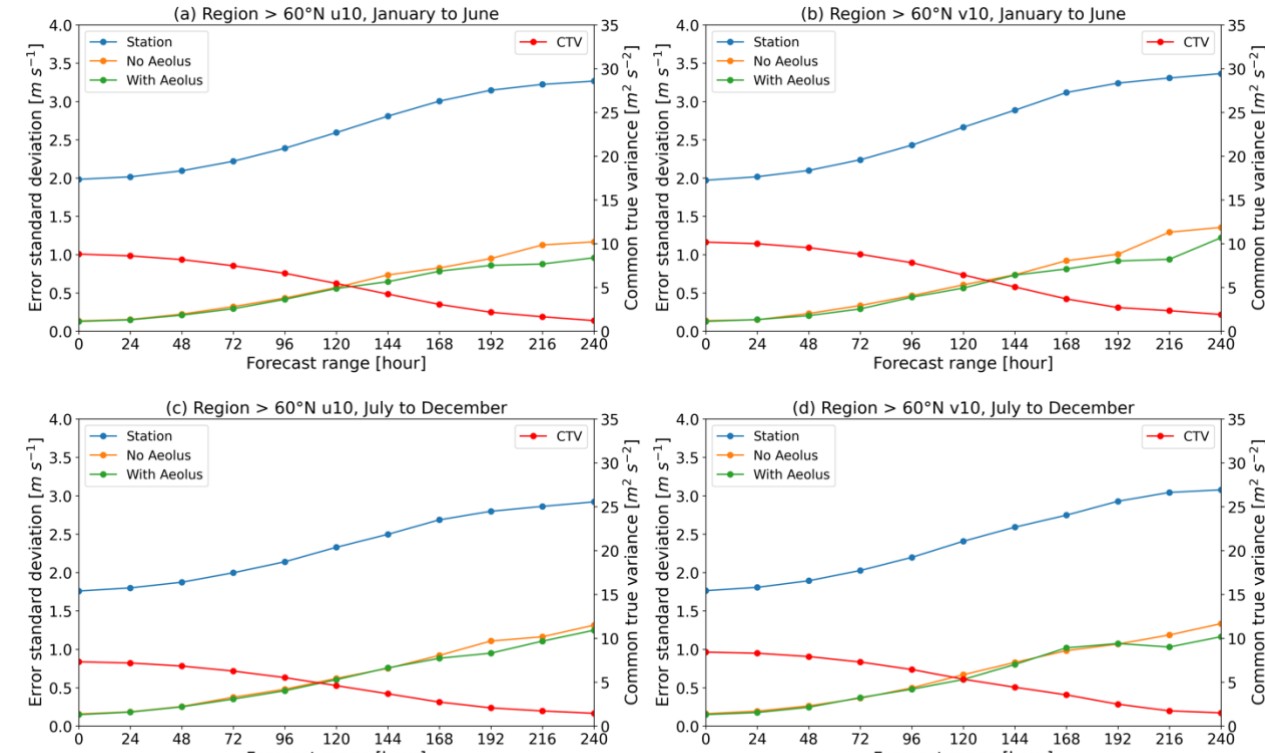

Figure 11: Same as Figure 10 but for the first (a and b) and the second half-year (c and d) of 2020.





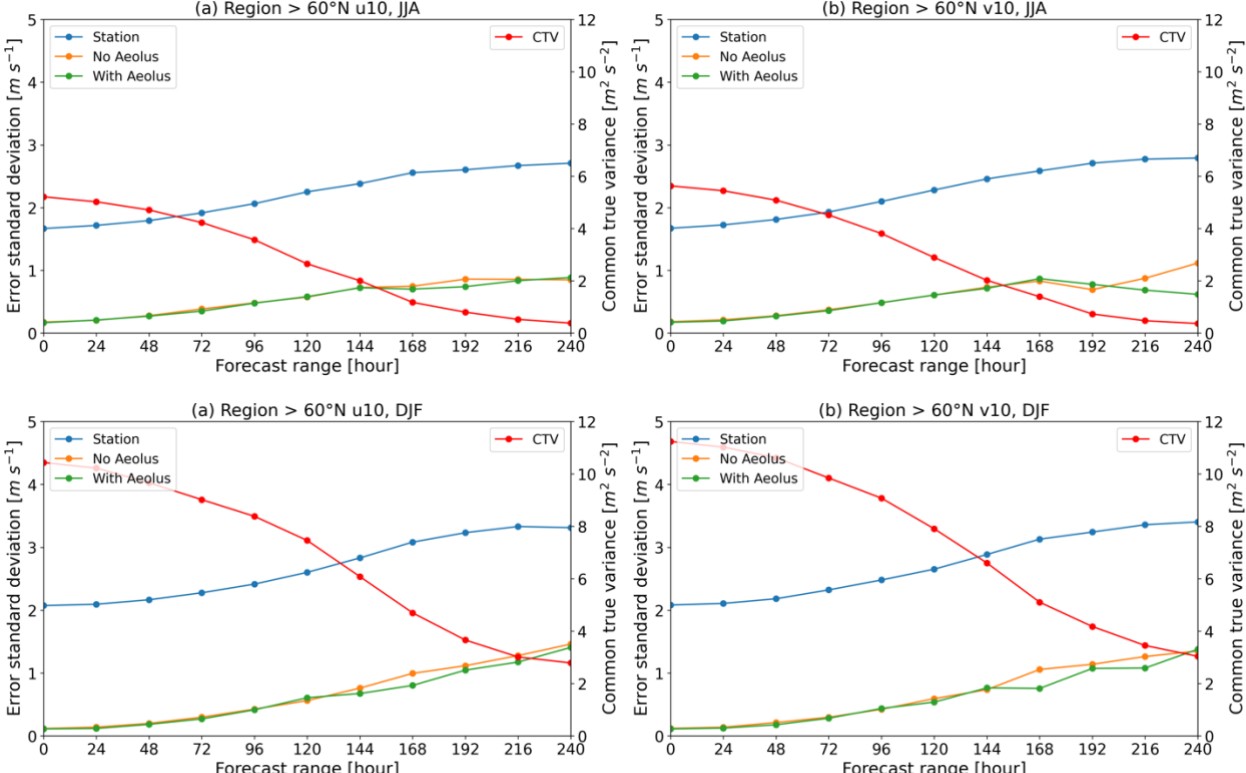

**Figure 12: Same as Fig. 10 but for boreal summer (a and b) and boreal winter (c and d), respectively.**

### 4.2.3 Correlation

The correlations between the two OSEs are higher for the u component and wind speed (R around 0.90) than for the v component (R around 0.6) over the high latitude region >60° N at T+120 h forecast step (Fig. 13). In contrast to the forecasts at the same hour in the Pacific (Fig. 6), the R-values are considerably lower in the northern high latitude region and the data clouds much more scattered in the high latitudes than in the tropical oceans.

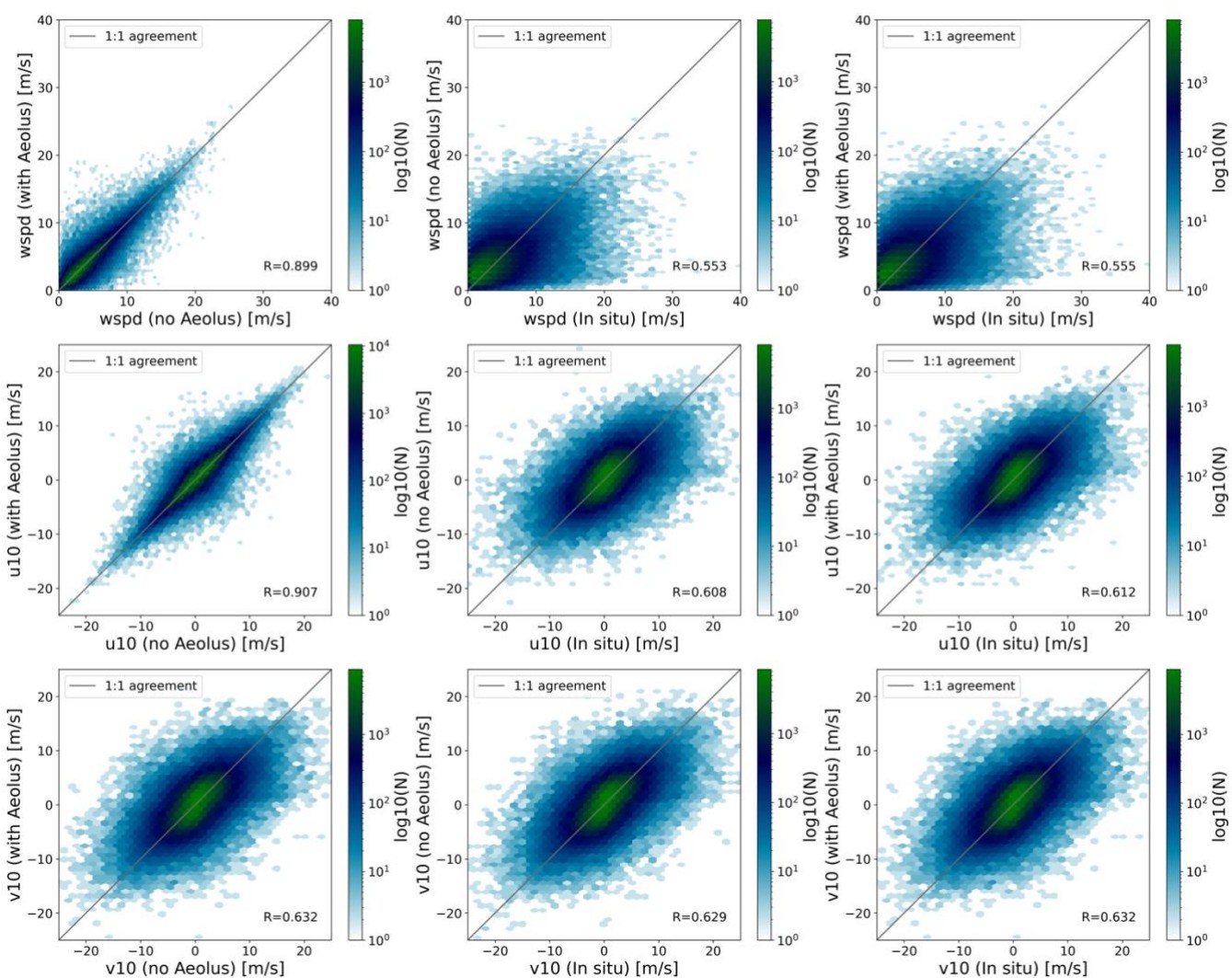

**Figure 13: Hexagonal binning plots of u, v components and wind speed at T+120 h for the region > 60° N for the year 2020 based on the ECMWF OSE forecast with and without Aeolus and weather station data. The colour of each hexagon indicates the number of samples in it.**

### 4.3 High latitude region in the Southern Hemisphere (> 60° S)

#### 4.3.1 Inter-comparison analysis

For the Southern Hemisphere, the impact of Aeolus on wind forecast is nearly neutral when considering the whole study period. The negative NDRMSEs were mainly found at T+96 h and +216 h (Fig. 14). More error reductions of u component and wind speed were found within T+96 h and at T+216 h and T+240 h during the first half-year of 2020 (Fig. 15). With





respect to the results for each season (Fig. 16), with the forecast range extending, there are more positive impacts of Aeolus on u component although these exist randomly on any season. Moreover, for the u component, the error reductions during winter months (June, July and August) are more evident than during summer months (January, February and December) at many forecast steps.

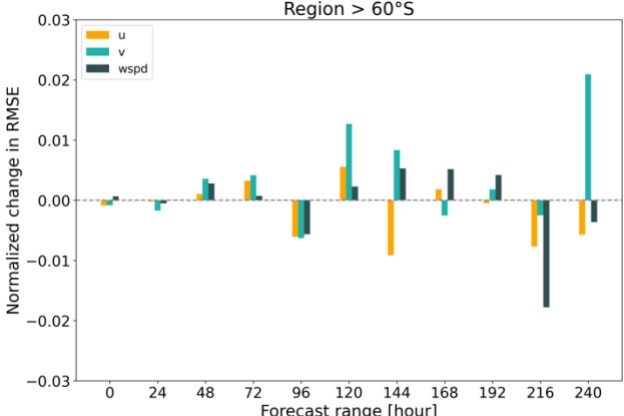

**Figure 14: Normalized difference in RMSE of u, v components and wind speed (wspd) as a function of forecast range for the region > 60° S for the year 2020 based on the ECMWF OSE forecasts with and without Aeolus and weather station data.**

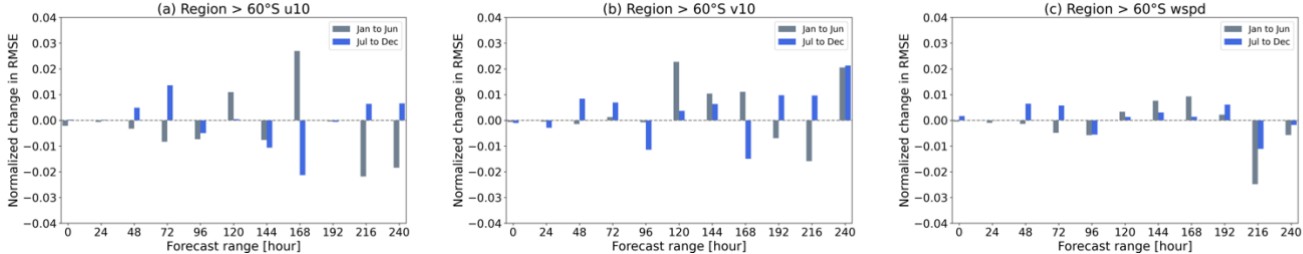

**Figure 15: Normalized difference in RMSE of u, v components as a function of forecast range for two different half years of 2020 for the region > 60° S based on the ECMWF OSE forecasts with and without Aeolus and weather station data.**



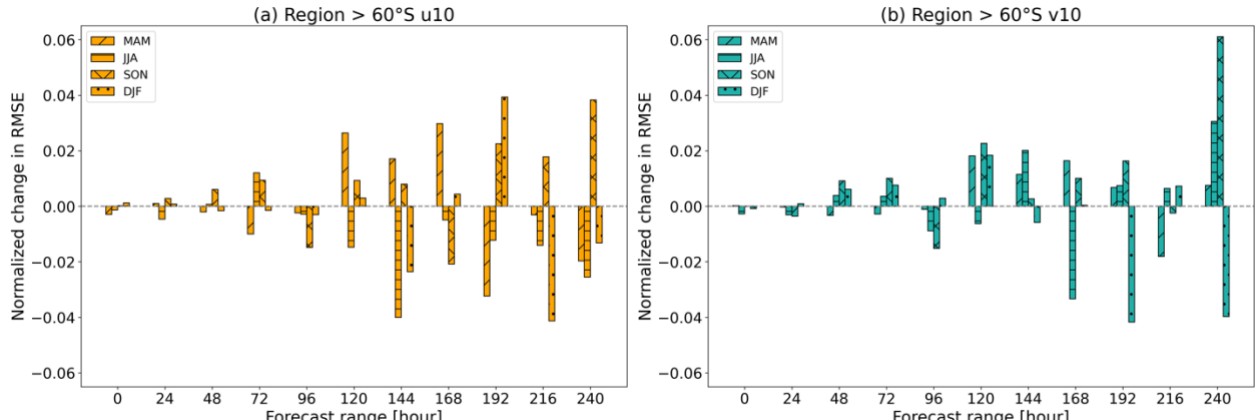


**Figure 16: Seasonal normalized difference in RMSE of u and v components as a function of forecast range for the region > 60° S for the year 2020 based on the ECMWF OSE forecasts with and without Aeolus and weather station data. MAM: March, April, and May; JJA: June, July, and August; SON: September, October, and November; DJF: December, January, and February.**

**4.3.2 Triple collocation analysis**

The results of triple collocation analysis are almost consistent with the results from intercomparison analysis. For the u components from OSEs (Fig. 17 (a)), the errors increase from around 0.3 m s⁻¹ at T+0 h to 2.5 m s⁻¹ at T+240 h with the values from the experiment with Aeolus wind assimilation slightly lower than the one without Aeolus up to T+96 h. The results of the v component have a similar pattern, but the improvement caused by Aeolus was mainly found at T+120 h and T+140 h (Fig. 17 (b)). In terms of the results for each half-year (Fig. 18), Aeolus has more positive impacts on the u 

component forecast with smaller errors within T+72 h and at T+216 h and T+240 h during the first half-year of 2020. Regarding the results for seasons, Aeolus improves the forecast more on the u component during winter months (June, July and August), while for the v component, the positive impacts were mainly found at the first T+48 h (Fig. 19 (a) and (b)). During the summer months (December, January, and February), the impact of Aeolus for wind forecast is nearly neutral except for the time steps from T+144 h for the v component.

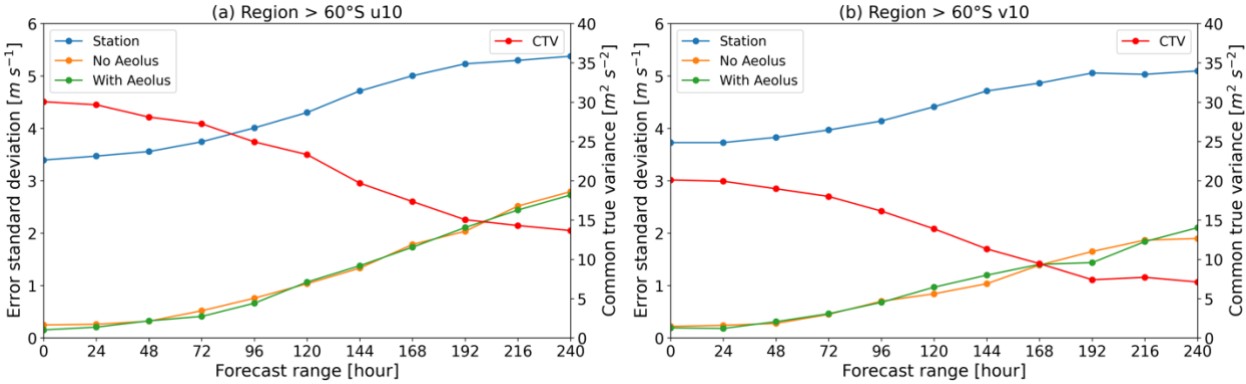


**Figure 17: Error standard deviation and common true variance of u and v components as a function of forecast range for the region > 60° S for the year 2020 based on the ECMWF OSE forecasts with and without Aeolus and weather station data.**





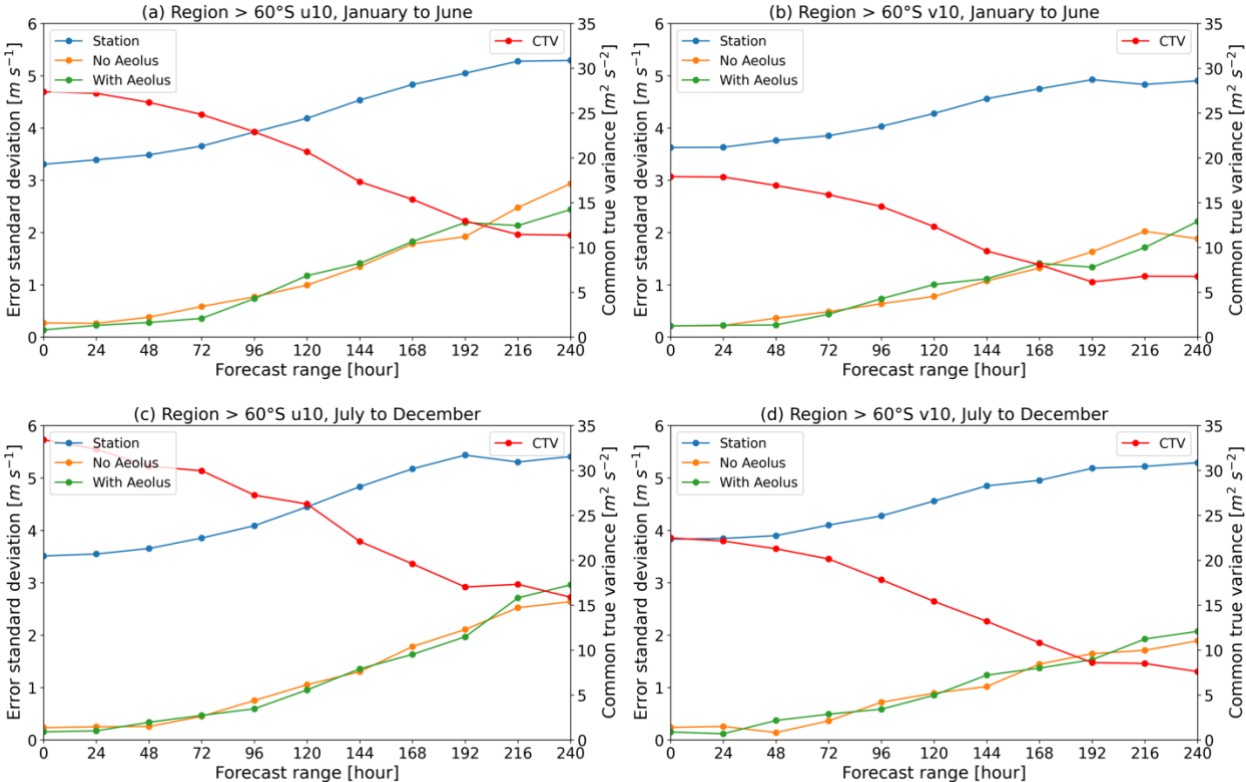

**Figure 18: Same as Fig. 17 but for the first (a and b) and the second half-year (c and d) of 2020.**





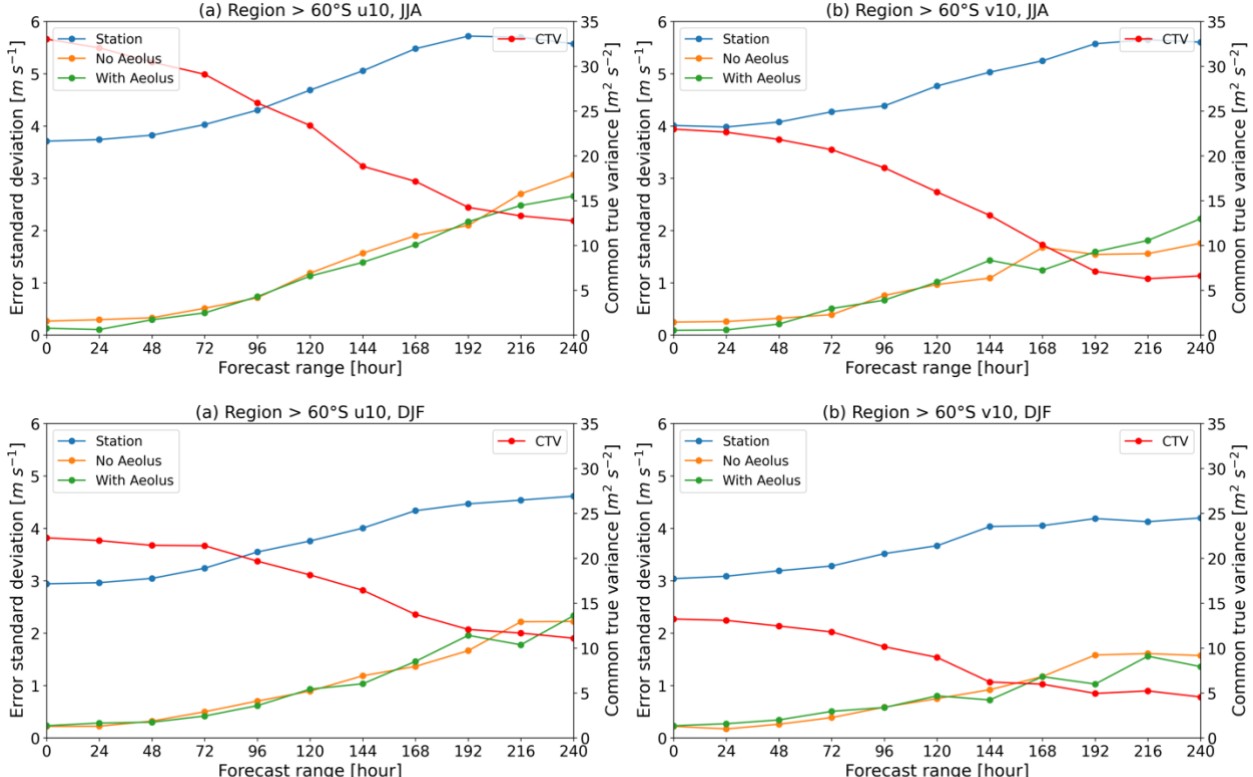

**Figure 19: Same as Fig. 17 but for the winter months (June, July and August) and summer months (December, January, and February) in the Southern Hemisphere, respectively.**

### 4.3.3 Correlation

Regarding the correlations, the wind speed and u components between the two OSEs with and without Aeolus assimilation are well correlated as the forecast extends, with R greater than 0.88 until T+120 h. For the u component and wind speed, the R values between OSEs and in situ measurements decrease gradually with forecast time, but the correlations are higher than for the v component. This is similar to the region > 60° N correlation results. The number of samples is much lower in the > 60° S region (N=12557) compared to > 60° N (N=229054) but is of the same order of magnitude as the samples in the Pacific (N=13389). However, the R-values between the region >60° S and the tropical Pacific differ greatly, with a much larger scatter and lower R-values in the region > 60° S.



Region > 60°S (+120 h, N=12557)



**Figure 20: Scatter plots of u, v components and wind speed at +120 h forecast for the region > 60° S for the year 2020 based on the ECMWF OSE forecasts with and without Aeolus and weather station data. The colour of each hexagon indicates the number of samples in it.**

## 5 Discussion

By taking in situ measurements as the reference, we evaluated the impact of Aeolus data assimilation on wind forecast at the surface level based on the ECMWF OSEs. According to the results from inter-comparison analyses for tropical oceans, there are positive impacts of Aeolus winds on sea surface wind forecast over all three ocean basins at T+48 h forecast steps, which





is roughly consistent with the verifications based on the model analysis at ECMWF (Rennie and Isaksen, 2022) and possibly due to the down propagation of Aeolus increments to the surface. From the results for the NH high latitude region, Aeolus makes more positive impacts than negative impacts on surface wind forecast except for T+168 h. These results are almost identical to the results from ECMWF verification (Rennie and Isaksen, 2022). However, for the SH high latitude region, the

findings of our study differ from the results from the ECMWF study. Positive impacts were only found at T+96 h and T+216 h in our study, while more noticeable positive impacts were found within T+96 h from the ECMWF. This difference may be due to the poor spatial coverage of reference data in our study for the region > 60° S. Additionally, in our study, the differences in the RMSEs between the control experiment and the experiment with Aeolus are not statistically significant at a significance level of 0.05 for most cases and time steps. We consider this in part to be due to the limited number of buoys

and weather stations distributed over the study regions. Another possible reason could be the representativeness of the point-based measurements compared to the coarse model resolution, which makes the errors between in situ measurements and model outputs large and random. This might be another reason for the difference between the RMSEs not being statistically significant.

To assess the impact of Aeolus data quality on its contribution to wind forecast, we also divided the study period into two half-year periods. There are more evident error reductions during the first half-year than during the second half-year for high-latitude regions, which suggests that the increasing random errors of Aeolus due to signal loss may degrade its impacts on wind forecast at the surface level. With respect to the impact of different seasons, the results show that Aeolus has a more positive impact on wind forecast during the winter months of each hemisphere than during the summer months. This is partly

attributed to the seasonal variation of solar background noise, which leads to larger random errors of Rayleigh-clear winds during summer months over polar regions and in the stratosphere (Reitebuch et al., 2022), thus resulting in larger forecast errors correspondingly.

The results from triple collocation analysis are approximately consistent with the results obtained from the inter-comparison

analysis, with lower error standard deviations corresponding to negative normalized changes in RMSE and indicating a positive impact from the Aeolus. For all cases, the in situ errors are always larger than the model errors since the result of triple collocation is with respect to the coarsest system (Vogelzang and Stoffelen, 2012), that is the ECMWF model resolution in this study. In addition, the error standard deviations of in situ measurements increase with forecast steps. This is partly because as the forecasts extend in time and become noisier, the common true variances (CTV) decrease as both the

experiments with Aeolus and no Aeolus winds fail to capture the smaller scale signals. Hence, the true small-scale signals measured by buoys or weather stations will go into the in situ noise (Ad Stoffelen, personal communication, 2022). This is why the in situ errors also grow with forecast time.





## 6 Conclusions

With the help of in situ measurements, the contribution of Aeolus wind assimilation to surface wind forecast was assessed
for tropical oceans (between 30° N and 30° S) and high latitude regions (> 60° N and > 60° S) through both inter-comparison
analysis and triple collocation analysis. The wind predictions come from the high resolution $T_{co}639$ OSEs with the ECMWF
model.

With Aeolus wind assimilation, a slight improvement of wind forecast is found over each tropical ocean basin, especially at
T+48 h. The decreasing quality of the DWL signal seems to have little impact on model performance in surface wind
forecast over the tropical ocean regions, except for the v component over the Atlantic. For the high-latitude region in the NH,
Aeolus has positive impacts for almost all forecast steps, and the positive influence becomes more evident with forecast
extending and especially for the first half-year of 2020 and for winter months owing to the better behaviour of the Aeolus.
Unlike the NH, the contribution of Aeolus to the high-latitude region in the SH is not that obvious. The error reduction is
mainly found for the u component and wind speed for the first half-year of 2020 compared to the second half-year.
Additionally, smaller errors for the u component are obtained during June, July and August for most forecast steps.
Moreover, the error standard deviations of u and v wind components from OSEs for tropical ocean basins and the region >
60° N remain within 1 m s$^{-1}$ with forecast extending, which are lower than the values for the high latitude region > 60° S that
grow from about 0.2 m s$^{-1}$ at T+0 h to about 2-3 m s$^{-1}$ at T+240 h.


Notwithstanding the limited spatial coverage of the reference data, the research findings of this study demonstrate the
potential of Aeolus observations on surface wind forecasts with the ECMWF model over the tropical ocean and the high
latitude regions. It also shows the prospects of the future spaceborne DWL programme for practical applications in wind-
related activities, such as ocean shipping and wind farm operation and maintenance.

**Appendix A**

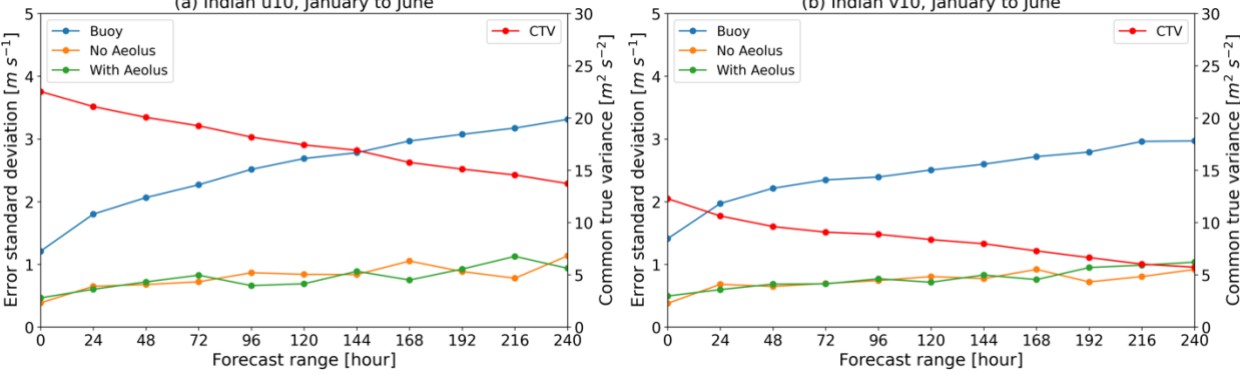





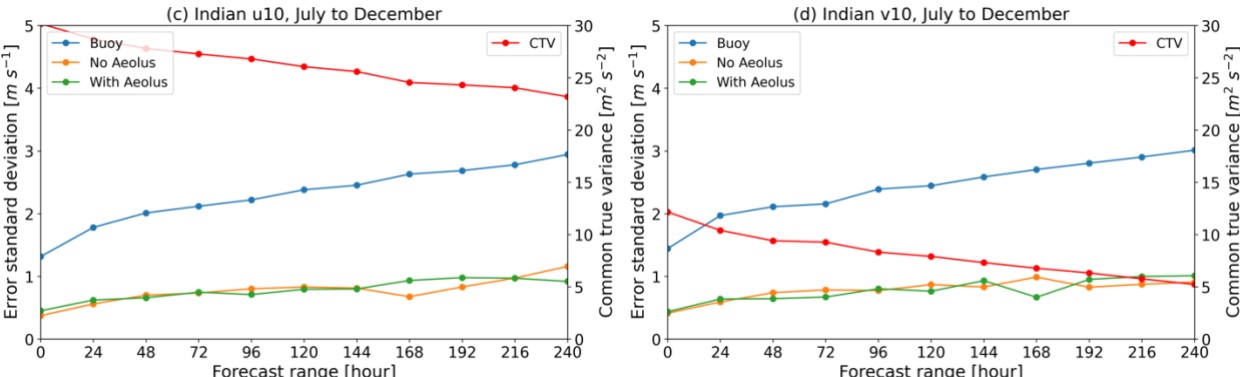

**Figure A 1: Error standard deviations and common true variances of u and v components as a function of forecast range for the Indian Ocean for the two half-year periods od 2020. January to June: (a) and (b); July to December: (c) and (d).**


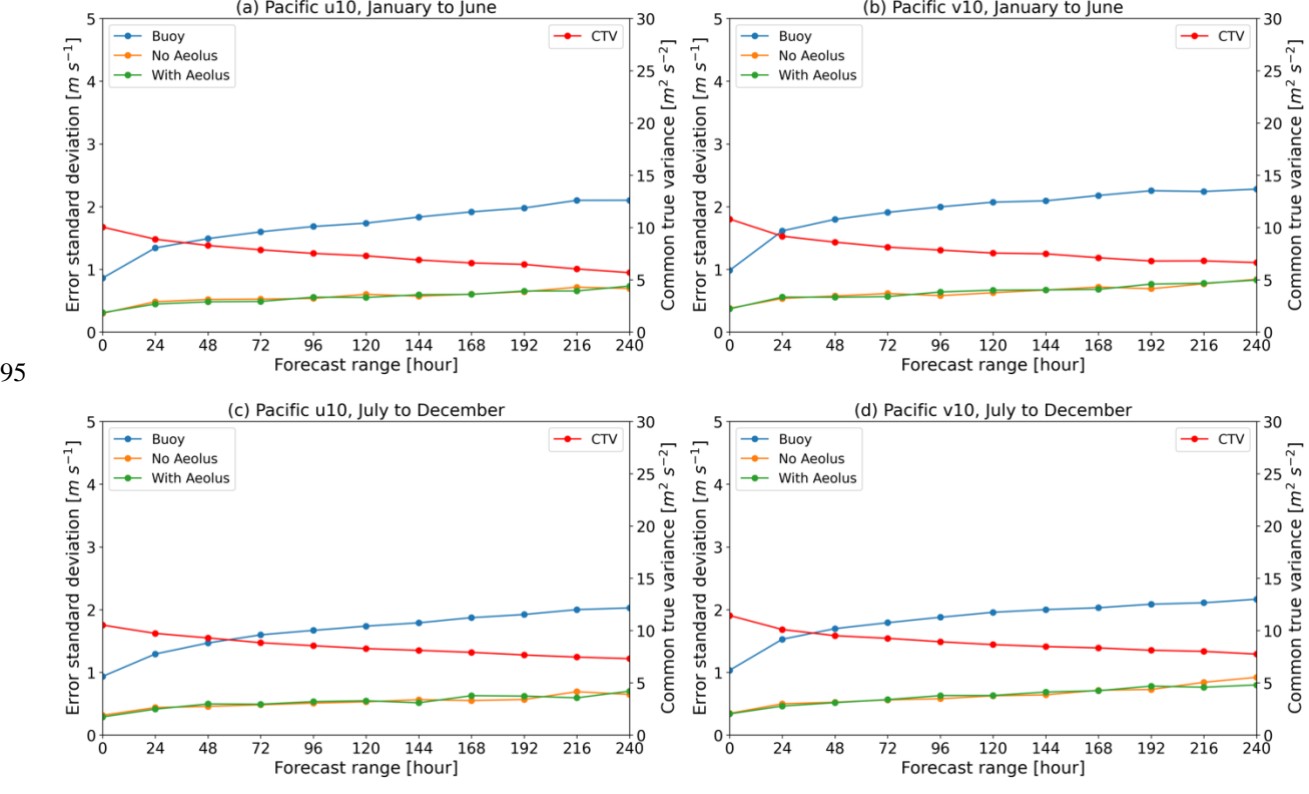

**Figure A 2: Same as Fig. A1 but for the tropical Pacific.**

*Data availability.* The OSEs were conducted by the ECMWF, and the u and v wind components were extracted from MARS (https://apps.ecmwf.int/mars-catalogue/, last access: 28 July 2022). The buoy measurements were obtained from Global Tropical Moored Buoy Array (https://www.pmel.noaa.gov/tao/drupal/disdel/, last access: 04 August 2022, National Oceanic and Atmospheric Administration Pacific Marine Environmental Laboratory). Wind information at weather stations is accessed via Global Hourly - Integrated Surface Database (https://www.ncei.noaa.gov/products/land-based-station/integrated-surface-database#:~:text=Global%20Climate%20Station%20Summaries%20Summaries%20are%20simple%20indicators,or%20longer%20time%20periods%20or%20for%20customized%20periods., last access: 11 August 2022, National Centers for Environmental Information).




*Author contributions.* HZ obtained the data, performed the data analysis, and drafted the manuscript. CH helped in interpreting the research findings. HZ revised the manuscript critically.

*Competing interests.* The authors declare that they have no conflict of interest.

*Acknowledgement.* This study is a part of PhD project Aeolus satellite lidar for wind mapping, a sub-project of the LIdar Knowledge Europe (LIKE) Innovative Training Network (ITN) Marie Skłodowska-Curie Actions funded by European Union Horizon 2020 (Grant number: 858358). We would like to thank the Royal Netherlands Meteorological Institute (KNMI) for being the secondment host institution. Our special appreciation goes to Ad Stoffelen from KNMI who gave us the idea to conduct this study and to Gert-Jan Marseille from KNMI for his assistance with OSE data retrieval. We would also like to show our gratitude to the ECMWF for conducting the OSEs and providing the data for analysis. We thank the National Oceanic and Atmospheric Administration Pacific Marine Environmental Laboratory for buoy data and the National Centers for Environmental Information for wind measurements at weather stations.

*Financial support.* This research is a part of the PhD project Aeolus Satellite Lidar for Wind Mapping, a sub-project of the Innovation Training Network Marie Skłodowska-Curie Actions: Lidar Knowledge Europe (LIKE) supported by the European Union Horizon 2020 (Grant number: 858358).

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
