# Peer review of "The impact of Aeolus winds on surface wind forecast over tropical ocean and high latitude regions"

_Atmospheric Measurement Techniques, 2022_

## Author Comment (AC1)

**Response to comment on amt-2022-311**

Anonymous Referee #1

Referee comment on "The impact of Aeolus winds on surface wind forecast over tropical ocean and high latitude regions" by Haichen Zuo and Charlotte Bay Hasager, Atmos. Meas. Tech. Discuss., https://doi.org/10.5194/amt-2022-311-RC1, 2022

We appreciate the anonymous referee's effort in providing valuable feedback on our manuscript. The insightful comments and suggestions are invaluable for improving our study and the manuscript. All questions and concerns have been addressed point-by-point with responses highlighted in blue, and the corresponding modifications (in orange) have been incorporated into the revised manuscript. The line numbers mentioned in this response letter correspond to the revised manuscript of tracked version.

Below are the responses to the comments and concerns from Referee #1.

In their manuscript "The impact of Aeolus winds on surface wind forecast over tropical ocean and high latitude regions", the authors analyze the impact of Aeolus data assimilation in the ECMWF model on surface wind in the tropics and high latitudes. This is done by comparing surface buoy and weather station data with two model runs. One with and one without assimilating Aeolus data. The comparison is done by means of normalized difference between the root mean square errors, triple collocation analysis and correlation analysis.

The paper in general is well structured and well written. Unfortunately, the performed analysis and interpretation has some deficiencies which need to be addressed before the publication can be recommended.

**Response:** Thank you very much for your positive feedback, and we will endeavour to address all deficiencies of our study.

The major point concerns the triple collocation. This method is usually used to assess the quality of three independent datasets. In the present study, the authors use it to compare two model runs from the same atmospheric model (which are not independent from each other) with an independent set of in-situ measurements. There is no explanation at all why this method should also be applicable to non-independent datasets and to forecasts.

**Response:** Thank you for pointing this out.

Triple collocation (TC) requires three independent data sets as the method is under the assumption that the errors of each two data sets are uncorrelated (Vogelzang and Stoffelen, 2012). Indeed, the two model runs are not fully independent of each other due to the same model formulation. However, as the forecast extends, the different initial conditions may lead to the random errors being more independent. Thus, to simplify TC equations, we assume that the errors from the two forecasts are independent.

We added this explanation in Section 2.1 (Sect.2.1) (Line: 155-157).

"Wind forecast errors from the two OSEs are considered uncorrelated because as the forecast extends, the different initial conditions may lead to the random errors being more

independent. Thus, to carry out TC analysis and simplify the equations, we assume that the errors from the two model runs are independent over the whole forecast range."

A second general point to explain would be, why the authors have used the relatively "coarse" ECMWF model for their analysis. They point out, that surface winds are especially important for ocean shipping and wind farming, which is why they focus on surface winds in tropical and polar regions. However, for these use cases often short-term forecasts and higher-resolved regional models are used. So why not perform the analysis on such model runs?

**Response:** Thank you for this question.

Our evaluation is based on the ECMWF model with an outer loop resolution of ~18 km in gird size. This is because the relatively low spatial (Rayleigh-clear winds: ~87 km; Mie-cloudy winds: ~12 km) and temporal resolution (7 days a repeat cycle) of Aeolus wind observations makes the global models are more likely to benefit from Aeolus data assimilation than high-resolution regional models (Hagelin et al., 2021; Mile et al., 2022; Rennie and Isaksen, 2022). Therefore, as a starting point, we would like to focus on the ECMWF model first. This will give us a better understanding of the influence of Aeolus on surface wind forecasts, which in turn guides us to apply Aeolus winds to regional models for practical applications in the future.

We have added the corresponding explanations in Sect. 1 (Line: 89-93).

"Due to the relatively low spatial and temporal resolution of Aeolus wind observations, global models are more likely to benefit from Aeolus data assimilation than high-resolution regional models (Hagelin et al., 2021; Mile et al., 2022; Rennie and Isaksen, 2022). Therefore, as a starting point, we would like to focus on the ECMWF model first. This will give us a better understanding of the influence of Aeolus on near-surface wind forecasts, which in turn guides us to apply Aeolus winds to regional models for practical applications."

As a third deficiency, I see the interpretation of the results from the tropical region. All comparison methods show a strongly varying impact over time. Nevertheless, the authors claim an improvement of the prediction, which from my point of view cannot be derived from the provided plots.

**Response:** Thank you very much for pointing this out.

We agree with this comment. Although there are negative values of the Normalized Change in RMSE at some forecast steps for tropical oceans, unfortunately, they are not statistically significant at a 95% confidence interval due to the limited number of data samples. It would be more appropriate to change the conclusion from the improvement of the prediction to limited impact.

We have revised the related text throughout the manuscript.

A detailed list specific comments is given below.

L64: Why are you are limiting your study to tropical and polar regions?

**Response:** This is because the tropical and polar regions are favourable to extreme weather but lack in situ measurements, and the model performance is usually not that good over these

regions, such as large bias over tropical regions (Sandu et al., 2020). Hence, we would like to investigate whether Aeolus data assimilation can contribute to near-surface wind forecast over these regions.

We have added the text in Sect. 1 (Line: 93-96).

"Considering tropical oceans and polar regions are favourable to extreme weather but lack in situ measurements and the model performance is usually not satisfactory in these regions, such as large bias over Inter-Tropical Convergence Zone (ITCZ) (Sandu et al., 2020), we would like to investigate whether the Aeolus can contribute to more reliable wind forecasts for these regions."

L70: I guess the datasets you are using are also included in the model reanalysis? So, what is the additional benefit of your study? The data assimilation in reanalysis at least somehow deals with the representativeness error between "large" scale model and point wise in situ measurement. You are barely touching this issue in your analysis.

**Response:** Thank you for these questions.

Indeed, the buoy data and weather station measurements are included in the model reanalysis. However, the model itself has large uncertainties especially over the tropical ocean regions partly due to the unresolved mesoscale convections (King et al., 2022), so we treat the in situ measurements as the truth rather than the model reanalysis to avoid this problem. Another benefit of our study is that we focus on wind components and wind speed instead of vector wind as the former is more relevant to practical applications.

In the revised manuscript, we have added the text regarding this issue in Sect.1 (Line: 98-103) and Sect.3.2 (Line: 274-275).

"Regarding the reference data set for evaluation, many verifications related to Aeolus OSEs were conducted by inter-comparing with model analysis that has global coverage and deals with the representation error between model scale and scales of observations (Garrett et al., 2022; Pourret et al., 2022; Rennie and Isaksen, 2022). However, there are fewer in situ measurements available over tropical and polar regions, and the mesoscale convections are not resolved well in the global NWP models, which leads to the large uncertainties of model analysis data in these regions (Sandu et al., 2020; King et al., 2022). Given this, taking in situ measurements as the reference can avoid this issue to some extent."

"We focus on error information of each wind component as well as wind speed instead of vector wind as the former is more relevant to practical applications."

L72: Here you state it yourself: triple collocation analysis is good for the evaluation of three independent data sets. Whereas in your study, I would not say that the two model runs are independent. So why can you use triple collocation?

**Response**: Thanks very much for this question.

Triple collocation (TC) analysis requires three independent data sets as the method is under the assumption that the errors of any two data sets are uncorrelated (Vogelzang and Stoffelen, 2012). Indeed, the two model runs are not fully independent of each other due to the same model formulation. However, as the forecast extends, the different initial conditions may lead

to the random errors being more independent. Thus, to simplify TC equations, we assume that the errors from the two forecasts are independent.

We added this explanation in Sect.2.1 (Line: 155-157).

"Wind forecast errors from the two OSEs are considered uncorrelated because as the forecast extends, the different initial conditions may lead to the random errors being more independent. Thus, to carry out TC analysis and simplify the equations, we assume that the errors from the two model runs are independent over the whole forecast range."

L75: Here you are talking about representative errors, whereas later on you state "the error covariances are free of representation error".

**Response**: Thanks for the comment. We apologize for the confusion related to representation errors.

The information here is for the general introduction of the triple collocation method, and the latter is for our case. The TC method uses the common signal of three systems, and the results are with respect to the system of the medium or lowest resolution of the three systems. When implementing the triple collocation method for three systems with different resolutions, such as high (buoy: point-based), medium (scatterometer: 25 km) and low (model:~150-200 km), we need to take representation error into account since the signal seen by buoys and the scatterometer can not be resolved by the model (Vogelzang and Stoffelen, 2012).

In our case, "the error covariances are free of representation error" is because the two model runs have the same spatial and temporal resolution. Thus, there is no common wind signal of in situ measurements and forecast (with/no Aeolus) not resolved by another forecast (no/with Aeolus). Hence, the representation errors do not play a role in our case.

We revised the corresponding text in Sect.1 (Line: 108-110) and Sect. 3.2 (Line: 256-262).

"Furthermore, TC analysis takes representation errors into account when three data sets have different spatial and temporal resolutions, and the result is with respect to the medium or lowest resolution among three systems."

"When implementing the triple collocation method for three systems with different resolutions, such as buoy measurements (point-based), scatterometer winds (25 km) and forecast winds from a global model (effective resolution: ~150-200 km), we need to take representation error into account since the model cannot resolve the signal seen by buoys and the scatterometers. In our case, two model runs have the same spatial and temporal resolution, as shown in Table 1. There is no common signal of System 1 (in situ measurements) and System 2 or 3 (control experiment or experiment with Aeolus data) not resolved by System 3 or 2 (experiment with Aeolus data or control experiment), which implies the representation errors of the error covariance are zero (i.e. $< e_1 e_2 > = < e_1 e_3 > = 0$). The forecast errors from the two model runs are assumed uncorrelated, thus $< e_2 e_3 > = 0$."

Section 3 – Method: The information provided in this section is very short. Could you please expand a bit more on the derivation of the different equations and especially the reasoning why you use two different methods and how they compare with each other. Especially regarding the triple collocation method, I would hope for a bit more explanation, why it can

be used under the current circumstances. As far as I know, the method usually uses three independent datasets. However, the two ECMWF model runs are by no means independent. Why do you still think the method is applicable and what does this mean for the equations. Do equations (5) – (7) till hold for the strongly correlated datasets 2 & 3?

**Response**: Thanks for these questions and suggestions.

We added more text to introduce the triple collocation method and three additional equations (Eq.(5)-(7) in the revised manuscript) to show how the error covariances affect the results (Ribal and Young, 2020). Since the step-by-step equation derivation has already been well documented in Vogelzang and Stoffelen (2012) and some other papers about the TC method (McColl et al., 2014), we only put the main equations in our paper to avoid too many complicated equations.

$$\sigma_1^2 = \langle e_1^2 \rangle = C_{11} - \frac{(C_{12} - < e_1 e_2 >)(C_{13} - < e_1 e_3 >)}{C_{23} - < e_2 e_3 >} \tag{1)(5}$$

$$\sigma_2^2 = \langle e_2^2 \rangle = C_{22} - \frac{(C_{12} - < e_1 e_2 >)(C_{23} - < e_2 e_3 >)}{C_{13} - < e_1 e_3 >} \tag{2)(6}$$

$$\sigma_3^2 = \langle e_3^2 \rangle = C_{33} - \frac{(C_{23} - < e_2 e_3 >)(C_{13} - < e_1 e_3 >)}{C_{12} - < e_1 e_2 >} \tag{3)(7}$$

where $C_{ii}$ is the variance of each system, and $C_{ij}$ is the covariance between the system i and j; and $\langle e_i e_j \rangle$ is the representation error of the error covariance between the system i and j. System 1: in situ measurements; System 2: forecast no Aeolus; System 3: forecast with Aeolus.

We use two different methods because we would like to explore whether the triple collocation method is applicable for model performance evaluation. For inter-comparison study, negative NCRMSEs indicate the relative error reductions owing to Aeolus data assimilation. The result of TC compares the random error of each system directly, with $\sigma_3$ smaller than $\sigma_2$ implying the positive impact of Aeolus, which corresponds to the negative NCRMSE if using inter-comparison analysis.

For implementing triple collocation, three independent data sets are required. In our study, as mentioned earlier, the wind forecasts from the two ECMWF model runs are assumed independent of each other because of different initial conditions that may lead to more independent random errors as the forecast extends. Due to the limited Aeolus winds assimilated below 850 hPa, the errors of wind forecasts at the initial stage may be correlated to some extent. This dependency may lead to $\langle e_2 e_3 \rangle \neq 0$. According to Eq.(1)-(3) (Eq.(5)-(7) in the revised manuscript), the random error of each system may be slightly over or under-estimated.

Since the objective of our study is to make an attempt to use the TC method for model performance assessment, and considering the dependence of the two model runs weakens with forecast extending, we assume no correlation in random errors between the two OSEs for all forecast steps.

We have added the above information to Sect.3.2 and Sect.5 (Line: 844-847).

L167: Are you now talking about triple collocation or NDRMSE? Which Figure are you describing? Please reference to make it easier for the reader to follow you.

**Response**: Thanks for this comment.

Here, we are talking about the NDRMSEs (NCRMSEs in the revised manuscript) from inter-comparison analyses for tropical oceans according to Fig. 3. We have added the figure numbers in Sect.4.1.1 (Line: 306).

"Figure 3 shows the NCRMSEs from inter-comparison analyses for three ocean basins."

In the revised manuscript, NDRMSE was changed to NCRMSE (Normalized change in RMSE) to make it consistent with the y-axis label of the plots.

L167 – 175: The results are strongly scattering and changing from positive to negative from time-step to time-step. So, I would be careful with saying the forecast is improved. Your analysis is not providing any reliable result here. Same on Figure 3, btw.

**Response**: Thank you for this comment. We have re-written Sect.4.1.1 (Line: 305-313).

"Figure 3 shows the NCRMSEs from inter-comparison analyses for three ocean basins. For the tropical Atlantic Ocean, the negative values are mainly found within T+72 h for the v vector and wind speed. The results for the tropical Indian Ocean do not show any trend in error reduction for wind components and wind speed. Compared to the tropical Atlantic Ocean and Indian Ocean, the Pacific witnesses negative values at more forecast steps, but the magnitude is weaker mainly within 1%. The negative values at T+48 h for both wind components and wind speed are common for the three ocean basins. Unfortunately, all the negative NCRMSEs are not statistically significant at the 95% confidence interval; thus, the overall impact of Aeolus on sea surface wind forecast is neutral for tropical regions. According to the results for two half-years (Fig.4), Aeolus data quality seems to have no influence on improving surface wind forecasts over the tropical ocean regions."

L196/197: As you write in L361f, this is expected. So please explain this to the reader already here.

**Response**: Thanks for this comment. We added more explanations in Sect. 4.1.2 (Line: 352-358)

"The error standard deviations are mainly within 1 m s$^{-1}$ from the two experiments, which are much lower than the buoy errors (> 1 m s$^{-1}$) for all three ocean basins since the true value of TC analysis is specified at the coarsest resolution among the three systems that is the model resolution in this study. The OSEs have an effective resolution of ~144 km (about 8 times the grid size) (Abdalla et al., 2013), which allows the model to capture large-scale atmospheric signals while small-scale details are lost. Consequently, errors from the two OSEs are smaller. Larger random errors of buoys are primarily due to the temporal and spatial representation errors related to collocation and the coarse model effective resolution."

L200: Again, I think it is problematic to talk about error reductions. The lines cross each other multiple times and there is no clear trend for one or the other.

**Response**: Thanks for this comment.

We have re-written the corresponding text. Please see Sect. 4.1.2 (Line: 358-361).

"Based on the results in Fig. 5, the impacts of Aeolus data assimilation on the forecasts for the tropical ocean basins are nearly neutral. In terms of the results for two half-year periods, there are no evident differences in error characteristics between the first and second half-year of 2020 for each ocean basin (Fig.6 and Appendix A)."

L220: "The correlations do not reveal much improvement in forecast skill between the two forecasts." I think this is the correct phrasing also for the two chapters before. No clear evidence for an improvement.

**Response**: Thank you for this comment.

We agree with that and have revised the corresponding text throughout the whole manuscript.

L232ff & L258ff: Any idea why? I guess somehow related to the storm season?

**Response:** Thank you for these questions.

Yes, our new results for different wind speed ranges for the region > 60° N demonstrate that Aeolus tends to reduce the forecast errors more for moderate-to-high winds (6-11 m s$^{-1}$) than for light winds (< 6 m s$^{-1}$), as shown in Fig.1 (Fig. 9 in the revised manuscript). Thus, during the stormy season, more error reductions are expected. Therefore, apart from the solar background issue, the different contributions of Aeolus winds under different wind speed ranges could be another reason for seasonal variation of error reductions.

We have added the new results in Sect.4.2.1 (Line: 453-458) and modified the discussions in Sect.5 (Line: 782-786).

"Regarding the results for different wind speed categories (Fig.9), the noticeable error reductions tend to exist earlier from T+96 h forecast step for moderate to fresh breeze (6 < wspd ≤11 m s$^{-1}$) compared to the light wind category; for the category of strong breeze to near gale (11 < wspd ≤17 m s$^{-1}$), the negative NCRMSEs for v component exist from the T+120 h forecast step; while the largest impact on u and v components are observed at T+216 h and T+192 h, respectively, when wind speeds greater than 17 m s$^{-1}$, but a further demonstration is required due to limited amount of data samples in this category (N: around 1200)."

"Another possible reason for the seasonal variation in error reduction is the different contributions of Aeolus data assimilation under different wind speed ranges. According to Fig.9, more error reductions are found when wind speeds are greater than 6 m s$^{-1}$ for the region > 60° N. Thus, during the stormy season, which is usually the wintertime for the high-latitude regions, there could be more evident error reductions."

[Figure]

Figure 1 (Figure 9). Normalized change in RMSE for u, v wind components and wind speed (wspd) for the region > 60° N for different wind speed ranges for the year 2020 based on ECMWF OSE forecasts with and without Aeolus against weather station measurements. Note that negative values indicate error reduction, implying the improvement in the forecast with Aeolus assimilation. (Same to Fig.8 but for different wind speed ranges.)

Figures 10 - 12: Can you elaborate a bit more on the temporal evolution of the CTV? Is this expected? Why?

**Response**: Thank you for this question.

As the forecasts extend, the common true variances are expected to decrease. This is because both the experiments with Aeolus and without Aeolus data assimilation are unable to capture the small-scale signals with forecast extending, thus resulting in decreasing CTVs.

We have integrated the explanations for the decreasing CTVs and growing in situ errors into the Sect.5 (Line: 848-852).

"This is partly because as the forecasts extend in time and become noisier, the common true variances decrease as both the experiments with Aeolus and no Aeolus winds are unable to capture the smaller scale signals. Hence, the true small-scale signals measured by buoys or weather stations will go into the in situ noise (Ad Stoffelen, personal communication, 2022). This is why the in situ errors also grow with forecast time."

L272f: This is a surprising result that the meridional wind with and without Aeolus assimilation are barely correlated anymore. Any idea why? At these high latitudes, the Aeolus measurement geometry starts to become favourable for measuring the meridional wind. Thus, I would expect a stronger impact on the meridional wind in this region compared to the

tropics. However, when looking at the in-situ data, there is only a marginal improvement. So, I do not really understand what is going on here.

**Response**: Thank you very much for pointing this out. We found a small error in our python script for plotting the meridional (v) winds from OSEs with and without Aeolus assimilation for high-latitude regions. After updating the plots, the R-values of the v component are comparable to the u component, with both being around 0.9 at the T+120 h forecast step.

Regarding the marginal improvement on the v component for the region > 60°N, one possible reason is that most weather stations in our study are located between 60°N and 75°N. At these latitudes, the Aeolus wind measurements are not fully meridional, so the contribution to the v component may be just slightly larger than to the u component. In addition, as shown from the inter-comparison analysis, Aeolus' impact on surface wind forecasts is quite small overall, although a slightly larger impact on v components is observed. Given these, the R values (forecast v.s.in situ) may not be able to reflect this small difference in the improvement between u and v components.

We have updated the Sect. 4.2.3 and Sect.4.3.3.

"Regarding the correlations for the region > 60° N, the wind components and wind speed between the two OSEs with and without Aeolus assimilation are well correlated as the forecast extends, with R-values greater than 0.90 until T+120 h (Fig.15 (a), (d) and (g)). Moreover, with the forecast extending, the R-values of the forecasts with Aeolus versus in situ measurements are slightly larger than the ones without Aeolus data, which is in line with the inter-comparison analysis, suggesting a minimal improvement in wind forecast. However, different from the inter-comparison analysis, the R-value is not sensitive to reflect which wind component can benefit more from Aeolus data assimilation."

"About the correlations for the region > 60° S, the wind components and wind speed between the two OSEs show strong agreement as the forecast extends, with R values consistently greater than 0.89 up to T+120 h (Fig.22 (a), (d) and (g)). This pattern is comparable with the results for the region > 60° N, although the number of data samples is much lower in the region > 60° S. Moreover, the R-values of each two systems decrease gradually with forecast time, but the correlations for the u and v components are stronger than those for the wind speed for all forecast steps. In addition, the correlations between the OSEs and the in situ measurements are consistent with the inter-comparison results, with R-values of the forecast with Aeolus versus in situ data higher than the ones without Aeolus corresponding to the negative NCRMSEs."

[Figure]

Figure 2 (Figure 15). Hexagonal binning plots of u, v components and wind speed at T+120 h for the region > 60° N for the year 2020 based on the ECMWF OSE forecasts with and without Aeolus and weather station data. The colour of each hexagon indicates the number of samples in it.

[Figure]

Figure 3 (Figure 22). Hexagonal binning plots of u, v components and wind speed at T+120 h for the region > 60°S for the year 2020 based on the ECMWF OSE forecast with and without Aeolus and weather station data. The colour of each hexagon indicates the number of samples in it.

L334ff: This might hold for this one time step, but as mentioned before this is strongly varying depending on forecast time and I would be careful with such a general statement.

**Response**: Thank you for this comment.

We have revised the corresponding text in Sect.5 (Line: 762-769).

"According to the results of inter-comparison analyses for tropical oceans, the impact of Aeolus on sea surface wind forecast is nearly neutral overall. However, negative NCRMSE values are observed across all three ocean basins at the T+48 h forecast step. Despite not being statistically significant, this result is consistent with the verifications based on the model analysis at ECMWF (Rennie and Isaksen, 2022), but further demonstration is required with more in situ measurements."

L342ff: Again, the question: What is the benefit of your study compared to the ECMWF study? It seems to me you have strong issues with the representativeness and the spatial coverage (and number if samples), which are not present in the ECMWF study.

**Response**: Thanks for this question.

ECMWF study takes operational analysis as the reference data set, but the model performance over tropical and polar regions is not good due to convections and/or lack of observations (Sandu et al., 2020; King et al., 2022). We use in situ measurements to avoid these issues. Another benefit of our study is that for the tropical region, we evaluated the model performance for each ocean basin separately, trying to provide error information geographically, while the study from ECMWF focuses on the zonal average. Apart from these, our assessment focuses on the u and v components as well as wind speed instead of vector wind, as the former is more relevant to real applications. Regarding representativeness and spatial coverage, these are indeed the drawbacks when taking in situ measurements being a reference.

We have added the corresponding information in Sect. 1 (Line: 100-103) and Sect. 3 (Line: 273-275).

"However, there are fewer in situ measurements available over tropical and polar regions, and the mesoscale convections are not resolved well in the global NWP models, which leads to the large uncertainties of model analysis data in these regions (Sandu et al., 2020; King et al., 2022). Given this, taking in situ measurements as the reference can avoid this issue to some extent."

"The analyses were performed for each ocean basin, regions > 60° N and > 60° S, respectively, aiming to provide error information geographically. We focus on error information of each wind component as well as wind speed instead of vector wind as the former is more relevant to practical applications."

L353ff: The solar background issue might be one reason. Another, physical reason might be related to the seasonality of the atmosphere. Winter is storm season in the high latitudes and correct wind information might have stronger impact on the long term in strong wind conditions. Thus, I would suggest to also look at different wind speed regimes before concluding this to be an instrument problem. Different wind speed regimes / synoptic situations might also be interesting especially for the use cases in shipping and energy production. Strong winds are the most problematic for these use cases. Thus, a more precise prediction of strong wind cases would be very beneficial.

**Response**: Thank you very much for this constructive suggestion.

We divided the collocated data samples into four categories based on the wind speeds of in situ measurements (Met Office, 2023), and for each category we evaluated the NCRMSE for u, v components and wind speed. The analysis is only for the region > 60° N owing to enough data samples. The results indicate that Aeolus tends to impact moderate-to-strong wind forecasts more.

These new results are added to the revised manuscript. Please see Sect.3 (Line: 297-298) and Sect. 4.2.1 (Line: 453-458).

"Moreover, for the region > 60° N, we divided the data samples into four categories based on the in situ wind speeds (Table 2) and investigated the impact of Aeolus under different wind speed ranges (Met Office, 2023)."

Table 1 (Table 2). Wind speed categories.

| Category | Wind speed range [m s⁻¹] | Description |
|:---:|:---:|:---:|
| **a** | wspd ⩽ 6.0 | Light air to gentle breeze |
| **b** | 6.0 < wspd ⩽ 11.0 | Moderate breeze to fresh breeze |
| **c** | 11.0 < wspd ⩽ 17.0 | Strong breeze to near gale |
| **d** | wspd > 17.0 | Gale to hurricane |

"Regarding the results for different wind speed categories (Fig.9), the noticeable error reductions tend to exist earlier from T+96 h forecast step for moderate to fresh breeze (6 < wspd ≤11 m s⁻¹) compared to the light wind category; for the category of strong breeze to near gale (11 < wspd ≤17 m s⁻¹), the negative NCRMSEs for v component exist from the T+120 h forecast step; while the largest impact on u and v components are observed at T+216 h and T+192 h, respectively, when wind speeds greater than 17 m s⁻¹, but a further demonstration is required due to limited amount of data samples in this category (N: around 1200)."

[Figure]

Figure 4 (Figure 9). Normalized change in RMSE for u, v wind components and wind speed (wspd) for the region > 60° N for different wind speed ranges for the year 2020 based on ECMWF OSE forecasts with and without Aeolus against weather station measurements. Note that negative values indicate error reduction, implying the improvement in the forecast with Aeolus assimilation. (Same to Fig.8 but for different wind speed ranges.)

L388f: Since future spaceborne DWL programmes have not been mentioned throughout the whole paper, I wonder if a statement on this topic is well funded here. If the authors want to comment on this topic, they should also include a statement on what would be necessary to further improve the surface wind forecasts they analyse (e.g. wind vector instead of HLOS, higher temporal and/or spatial resolution). I would recommend to completely remove this sentence.

**Response**: Thank you very much for this suggestion. We have removed this sentence in the revised manuscript.

**Additional clarifications:**

In addition to addressing all concerns from the anonymous reviewers, we corrected a small error in data quality control for high-latitude regions and re-plotted all related figures. For TC analyses, we adjusted the method and re-processed the data without removing the outliers in order to make the results reflect the real forecast errors. Additionally, the normalized difference in root-mean-square error (NDRMSE) was changed to the normalized change in root-mean-square error (NCRMSE) to be consistent with the y-axis label of the plots for inter-comparison analysis.

**References:**

Abdalla, S., Isaksen, L., Janssen, P. A. E. M., and Wedi, N.: Effective spectral resolution of ECMWF atmospheric forecast models, ECMWF, Newsletter Number 137, 19–22, doi:10.21957/rue4o7ac, 2013.

Hagelin, S., Azad, R., Lindskog, M., Schyberg, H., and Körnich, H.: Evaluating the use of Aeolus satellite observations in the regional numerical weather prediction (NWP) model Harmonie–Arome, Atmos. Meas. Tech., 14, 5925–5938, https://doi.org/10.5194/amt-14-5925-2021, 2021.

King, G. P., Portabella, M., Lin, W., and Stoffelen, A.: Correlating extremes in wind divergence with extremes in rain over the Tropical Atlantic, https://mdc.coaps.fsu.edu/scatterometry/meeting/docs/2022/King-IOVWST-2022.pdf (last access: 06 March 2022), 2023.

McColl, K. A., Vogelzang, J., Konings, A. G., Entekhabi, D., Piles, M., and Stoffelen, A.: Extended triple collocation: Estimating errors and correlation coefficients with respect to an unknown target: EXTENDED TRIPLE COLLOCATION, Geophys. Res. Lett., 41, 6229–6236, https://doi.org/10.1002/2014GL061322, 2014.

Met Office: Beaufort wind force scale: https://www.metoffice.gov.uk/weather/guides/coast-and-sea/beaufort-scale, last access: 24 February 2023.

Mile, M., Azad, R., and Marseille, G.: Assimilation of Aeolus Rayleigh-Clear Winds Using a Footprint Operator in AROME-Arctic Mesoscale Model, Geophysical Research Letters, 49, 1–11, https://doi.org/10.1029/2021GL097615, 2022.

Rennie, M. and Isaksen, L.: The NWP impact of Aeolus Level-2B winds at ECMWF, ECMWF, 227 pp., https://confluence.ecmwf.int/display/AEOL/L2B+team+technical+reports+and+relevant+papers?pre

view=/46596815/288355970/AED-TN-ECMWF-NWP-025--20220810_v5.0.pdf (last access: 20 October 2022), 2022.

Ribal, A. and Young, I. R.: Global Calibration and Error Estimation of Altimeter, Scatterometer, and Radiometer Wind Speed Using Triple Collocation, Remote Sens., 12, 1997, https://doi.org/10.3390/rs12121997, 2020.

Sandu, I., Bechtold, P., Nuijens, L., Beljaars, A., and Brown, A.: On the causes of systematic forecast biases in near-surface wind direction over the oceans, ECMWF, 21 pp., https://www.ecmwf.int/sites/default/files/elibrary/2020/19545-causes-systematic-forecast-biases-near-surface-wind-direction-over-oceans.pdf (last access: 22 February 2023), 2020.

Vogelzang, J. and Stoffelen, A.: Triple collocation, Royal Netherlands Meteorological Institute, 22 pp., https://cdn.knmi.nl/system/data_center_publications/files/000/068/914/original/triplecollocation_nwpsaf_tr_kn_021_v1.0.pd f?1495621500 (last access: 27 January 2022), 2012.

---

## Author Comment (AC2)

**Response to comment on amt-2022-311**

Anonymous Referee #2

Referee comment on "The impact of Aeolus winds on surface wind forecast over tropical ocean and high latitude regions" by Haichen Zuo and Charlotte Bay Hasager, Atmos. Meas. Tech. Discuss., https://doi.org/10.5194/amt-2022-311-RC2, 2023

We are grateful to the anonymous referee for taking time to review our manuscript and provide constructive comments. We have considered all comments and suggestions very carefully and revised the manuscript thoroughly.

In this response letter, all comments and concerns are addressed point-by-point, with responses highlighted in blue and the corresponding modifications in the revised manuscript in orange. The line numbers correspond to the revised manuscript of tracked version.

Below are the responses to the comments and concerns from Referee #2.

**General comments:**

This study examined the impact of Aeolus winds on surface wind forecast from the OSEs using ECMWF model. In general, the impact is quite small and not statistically significant at least in the tropical regions. The impact in the NH is negligible at forecast lead times < 192h. The triple colocation analysis results look very noisy and hard to interpret. It is not clear how the correlation between the OSEs will help to justify the performance of the OSEs. In summary, I do not see any significant impact of Aeolus winds on the forecast of surface winds from the OSEs.

**Response**: Thank you very much for these comments.

For the tropical ocean regions, the impact of Aeolus is indeed quite small and insignificant, although we found some negative values of normalized change root-mean-square errors (NCRMSE) within the T+144 h for the Atlantic and Pacific regions. We have revised the corresponding statements and corrected the conclusions.

For the high-latitude region in the Northern Hemisphere, the noticeable impact is found mainly from T+192 h onward, which is possibly owing to the downward propagation of Aeolus increments to the surface since there are a limited number of low-level (> 850 hPa) Aeolus winds inland assimilated into the model (Fig. 1).

To facilitate the interpretation of triple colocation results, we quantified the uncertainties of forecast errors at a 95% confidence interval by using the bootstrap method for each case and updated all figures.

Regarding the correlation coefficients between the OSEs and in situ measurements, we apologize for the unclear statements. More justifications have been added to each related section.

Thanks very much again for the feedback. We have tried our best to improve the manuscript.

**Specific comments:**

Abstract, line 13: It is not clear how do you get this conclusion: "The results show that with Aeolus data assimilation, the tropical sea surface wind forecast could be slightly improved at some forecast time steps." This is the opposite to the statement from line 175: "Unfortunately, the NDRMSEs are not statistically significant at a 95% confidence interval for all three tropical ocean regions."

**Response**: Thank you very much for pointing this out. We apologize for this conflicting statement.

Although there are negative NCRMSEs at some forecast steps, they have large uncertainties due to the limited number of data samples. We think it would be more appropriate to change the statement in the Abstract to "The results of the inter-comparison analysis show that Aeolus data assimilation has a limited impact on sea surface wind forecasts for tropical regions when compared with buoy measurements." (Line: 12-14)

Section 2.1: The resolution of the ECMWF model version is different from that of Rennie et al (2022). Have you re-tuned the specified observational error for Rayleigh and Mie winds for these specific OSEs in this study? It will be helpful to add some information about how many Aeolus Rayleigh and Mie winds are assimilated into the OSEs in the lower troposphere.

**Response**: Thanks very much for this question and suggestion.

I guess the ECMWF model version you mentioned is from the paper by Rennie et al. (2021). Their study is based on the model version CY46R1.2 for the early FM-B period and CY47R1.1 for the Mid-2020 period with an outer loop resolution of $T_{co}399$ (about 29 km grid). Our study is based on the CY47R2 with an outer loop resolution of $T_{co}639$ (about 18 km grid) for FM-B period using 2[nd] reprocessed data set, but the quality control decisions applied to this OSE are the same as the early FM-B period OSE. For the lower troposphere (> 850 hPa), only Mie-cloudy winds with estimated errors smaller than 5 m s$^{-1}$ are assimilated. Detailed information on quality control decisions for each OSE is documented in Rennie and Isaksen (2022).

We also created a map showing the averaged number of Mie-cloudy winds assimilated into the model per cycle below 850 hPa (Fig.1). More low-level Aeolus winds are assimilated over the ocean regions than inland regions and over low-to-mid-latitude regions than high-latitude regions.

The map and corresponding information have been added to the Sect. 2.1 (Line: 148-153).

"For the lower troposphere (> 850 hPa), only Mie-cloudy winds with an estimated error smaller than 5 m s$^{-1}$ were assimilated into the model. Detailed information on quality control decisions for the OSEs is documented in Rennie and Isaksen (2022). Figure 1 illustrates the geographical distribution of the averaged number of L2B Mie-cloudy winds assimilated per cycle below 850 hPa. More low-level Aeolus winds are assimilated over the ocean regions than inland regions and over low-to-mid-latitude regions than high-latitude regions."

[Figure]

Figure 1 (Figure 1). Map of the averaged number of L2B Mie-cloudy winds at pressure > 850 hPa assimilated into the model.

Line 120: Please explain more why those stations with weak correlations (R < 0.5) with the analysis should be removed. Add numbers of stations were removed.

**Response**: Thanks very much for this comment.

One reason is that when the weak correlations are caused by very limited data samples during the study period, such as due to freeze or instrument malfunction, we consider the data quality of those available samples are still questionable. Another reason is that the weak correlations may imply the limited spatial representativeness of those stations, especially over the complex terrain. After quality control, there are 751 (223) and 56 (30) stations available (removed) over the high latitude regions in the Northern and Southern Hemisphere, respectively.

We have added the explanation in Sect. 2.3 (Line: 187-191).

"One reason is that when the poor correlations are caused by very limited data samples during the study period, such as due to freeze or instrument malfunction, we consider the data quality of those available samples are still questionable. Another reason is that the weak correlations may imply the limited spatial representativeness of those stations, especially over the complex terrain. After quality control, there are 751 (223) and 56 (30) stations available (removed) over the high-latitude regions in the Northern and Southern Hemisphere, respectively (Fig. 2)."

Line 199, fig. 4: The triple collocation analyses of the two OSEs (no and with Aeolus) show no evident difference to me. Are the differences are statistically significant? This also applies to Figs. 10, 11.12, 17,18,19.

**Response**: Thanks very much for this question.

For the forecast errors derived from the triple collocation method, we quantified their uncertainties at the 95% confidence interval by using the bootstrap method. All corresponding figures have been updated in the revised manuscript.

Here, we show the updated figures for the tropical ocean basins, region > 60° N and region > 60°S for the year 2020 as examples. It can be seen that significant error reductions are found for the region > 60°N, particularly after T+168 h, while for other regions the errors have large uncertainties.

The corresponding text was also added in the Sect.4.1.2, Sect.4.2.2 and Sect.4.3.2.

[Figure]

[Figure]

Figure 2 (Figure 5). Error standard deviation and common true variance (CTV) of u and v wind components from triple collocation for the tropical Atlantic Ocean (a and b), Indian Ocean (c and d) and Pacific Ocean (e and f) for the year of 2020 based on the ECMWF OSE forecasts with and without Aeolus data assimilation and buoy data.

[Figure]

Figure 3 (Figure 12). Error standard deviation and common true variance (CTV) of u and v components as a function of forecast range for the region > 60° N for the year 2020 based on the ECMWF OSE forecast with and without Aeolus and weather station data.

[Figure]

Figure 4 (Figure 19). Error standard deviation and common true variance of u and v components as a function of forecast range for the region > 60° S for the year 2020 based on the ECMWF OSE forecasts with and without Aeolus and weather station data.

Line 200: The errors from the two OSEs are <1.0 m. Can you explain more why the errors are so small, considering a typical error of ~ 1.4 m/s of radiosonde near the surface.

**Response**: Thanks very much for this question.

The results of TC are with respect to the coarsest resolution among the three systems (Vogelzang and Stoffelen, 2012), which is the model resolution in this study. The effective

resolution of the model is about 8 times the grid size (Abdalla et al., 2013), so the effective resolution of the two OSEs is about 144 km. The model can capture the large-scale signal of the atmosphere but lose the small details, so the errors from the two OSEs are small. For the in situ measurements, such as buoys or radiosonde which can detect the small-scale signal of the atmosphere, the large random errors are primarily caused by the temporal and spatial representation errors associated with the collocation criteria and the coarse model effective resolution.

Another possible reason for the small forecast errors is that the wind forecasts for the first few days may not be fully independent due to the limited number of Aeolus low-level winds assimilated into the model, which leads to the $\langle e_2 e_3 \rangle \neq 0$. According to Eq.(5)-(7) in the revised manuscript (Ribal and Young, 2020), the error standard deviations for the OSEs may be underestimated, while those for the in situ measurements may be overestimated.

$$\sigma_1^2 = \langle e_1^2 \rangle = C_{11} - \frac{(C_{12} - <e_1 e_2>)(C_{13} - <e_1 e_3>)}{C_{23} - <e_2 e_3>} \qquad (1)(5)$$

$$\sigma_2^2 = \langle e_2^2 \rangle = C_{22} - \frac{(C_{12} - <e_1 e_2>)(C_{23} - <e_2 e_3>)}{C_{13} - <e_1 e_3>} \qquad (2)(6)$$

$$\sigma_3^2 = \langle e_3^2 \rangle = C_{33} - \frac{(C_{23} - <e_2 e_3>)(C_{13} - <e_1 e_3>)}{C_{12} - <e_1 e_2>} \qquad (3)(7)$$

where $C_{ii}$ is the variance of each system, and $C_{ij}$ is the covariance between the system i and j; and $\langle e_i e_j \rangle$ is the representation error of the error covariance between the system i and j. System 1: in situ measurements; System 2: forecast no Aeolus; System 3: forecast with Aeolus.

We have added more explanations in the Sect.5 (Line: 837-847).

Line 220: It is not clear how "The correlations can reveal improvement in forecast skill between the two forecasts?" Please explain.

**Response**: Thanks very much for this question.

Our hypothesis is that with Aeolus data assimilation, the plots of forecasts against in situ measurements should become less noisy compared to the ones without Aeolus. In other words, the correlations between the forecasts with Aeolus data assimilated and the in situ measurements should be stronger than the ones without Aeolus. In the manuscript, we wrote "The correlations do not reveal much improvement in forecast skill between the two forecasts" because there is not much increase in the correlation coefficients (R). Taking the T+120 h result of the tropical Pacific as an example, the R-values for the u component are around 0.81 for the forecasts (with/without Aeolus data) versus buoy data, and for the v component the R values are about 0.80.

We have added more explanations in Sect.4.1.3 (Line: 422-424). In addition, we corrected a small error in our python script for hexagonal binning plots for high-latitude regions, and all related figures and text have been updated. See Sect.4.2.3 and Sect.4.3.3

"The R-values for the u component are around 0.81 for the forecasts (with/without Aeolus data) versus buoy data (Fig. 7 (e) and (f)), and for the v component the R-values are about 0.80 (Fig. 7 (h) and (i)), which indicates there is almost no increase in correlation after assimilating Aeolus winds."

Pacific (T+120 h, N=13389)

[Figure]

Figure 5 (Figure 7). Hexagonal binning plots of u, v components and wind speed (wspd) at T+120 hour forecast for the tropical Pacific for the year 2020 based on ECMWF OSE forecasts with and without Aeolus and buoy data. The colour of each hexagon indicates the number of samples in it.

"Regarding the correlations for the region > 60° N, the wind components and wind speed between the two OSEs with and without Aeolus assimilation are well correlated as the forecast extends, with R-values greater than 0.90 until T+120 h (Fig.15 (a), (d) and (g)). Moreover, with the forecast extending, the R-values of the forecasts with Aeolus versus in situ measurements are slightly larger than the ones without Aeolus data, which is in line with the inter-comparison analysis, suggesting a minimal improvement in wind forecast. However, different from the inter-comparison analysis, the R-value is not sensitive to reflect which wind component can benefit more from Aeolus data assimilation."

[Figure]

Figure 6 (Figure 15). Hexagonal binning plots of u, v components and wind speed at T+120 h for the region > 60° N for the year 2020 based on the ECMWF OSE forecast with and without Aeolus and weather station data. The colour of each hexagon indicates the number of samples in it.

"About the correlations for the region > 60° S, the wind components and wind speed between the two OSEs show strong agreement as the forecast extends, with R values consistently greater than 0.89 up to T+120 h (Fig.22 (a), (d) and (g)). This pattern is comparable with the results for the region > 60° N, although the number of data samples is much lower in the region > 60° S. Moreover, the R-values of each two systems decrease gradually with forecast time, but the correlations for the u and v components are stronger than those for the wind speed for all forecast steps. In addition, the correlations between the OSEs and the in situ measurements are consistent with the inter-comparison results, with R-values of the forecast with Aeolus versus in situ data higher than the ones without Aeolus corresponding to the negative NCRMSEs."

[Figure]

Figure 7 (Figure 22). Scatter plots of u, v components and wind speed at +120 h forecast for the region > 60° S for the year 2020 based on the ECMWF OSE forecasts with and without Aeolus and weather station data. The colour of each hexagon indicates the number of samples in it.

Line 229: Are the positive impact statistically significant? Also applies to Fig. 14.

**Response**: Thanks very much for this question.

For Fig.7 (Fig. 8 in the revised manuscript), the positive impact is statistically significant at T+120 h, +216 h and 240 h for the u component, from T+192 h for the v component, and at T+192 h and T+216 h for wind speed.

For Fig.14 (Fig. 16 in the revised manuscript), no significant error reduction is found except for the wind speed forecast at T+216 h.

We have added the information of significance in Sect.4.2.1 (Line: 451-452) and Sect.4.3.1 (Line: 612-613).

"The significant positive impact is found at T+120 h, +216 h and 240 h for the u component, from T+192 h for the v component, and at T+192 h and T+216 h for wind speed."

"The negative NCRMSEs were mainly found at T+96 h and +216 h, but the significant error reduction is only at T+216 h for wind speed forecast (Fig. 16)."

Lines 232: The seasonal variations of error reductions may not necessarily solely due to the quality of Aeolus winds. Other factors may also contribute to this.

**Response**: Thank you for this comment.

Apart from the quality of Aeolus winds, model performance may vary depending on wind speeds, which may also contribute to the seasonal variations of error reductions. According to the new results for different wind speed ranges (Fig.9 in the revised manuscript), Aeolus data assimilation can lead to more error reductions when wind speed is greater than 6 m s$^{-1}$. Thus, there could be more evident error reductions during the stormy season, which is usually the wintertime of the high latitude regions.

We have added the new results in Sect.4.2.1 (Line: 453-458) and revised the discussion part in Sect.5 (Line: 782-786).

"Regarding the results for different wind speed categories (Fig.9), the noticeable error reductions tend to exist earlier from T+96 h forecast step for moderate to fresh breeze (6 < wspd ≤11 m s$^{-1}$) compared to the light wind category; for the category of strong breeze to near gale (11 < wspd ≤17 m s$^{-1}$), the negative NCRMSEs for v component exist from the T+120 h forecast step; while the largest impact on u and v components are observed at T+216 h and T+192 h, respectively, when wind speeds greater than 17 m s$^{-1}$, but a further demonstration is required due to limited amount of data samples in this category (N: around 1200)."

"Another possible reason for the seasonal variation in error reduction is the different contributions of Aeolus data assimilation under different wind speed ranges. According to Fig.9, more error reductions are found when wind speeds are greater than 6 m s$^{-1}$ for the region > 60° N. Thus, during the stormy season, which is usually the wintertime for the high-latitude regions, there could be more evident error reductions."

[Figure]

[Figure]

[Figure]

[Figure]

Figure 8 (Figure 9). Normalized change in RMSE for u, v wind components and wind speed (wspd) for the region > 60° N for different wind speed ranges for the year 2020 based on ECMWF OSE forecasts with and without Aeolus against weather station measurements. Note that negative values indicate error reduction, implying the improvement in the forecast with Aeolus assimilation. (Same to Fig.8 but for different wind speed ranges.)

Line 252: why the initial error from OSE with Aeolus is so small, only ~ 0.2 m/s?

**Response**: Thank you for this question. The answer is the same as the former question.

One reason is that the results of TC are with respect to the coarsest resolution that is the model resolution in our study (~144 km). The ECMWF model can capture large-scale signals and lack small details, so the errors from the two OSEs are really small. Another possible reason is that the wind forecasts for the first few days may not be fully independent due to the limited number of Aeolus low-level winds assimilated into the model; thus, the model errors might be underestimated.

We have added the explanation in the Sect.5 (Line: 837-847).

Lines 335, 387: This is the opposite to the statement from line 175: "Unfortunately, the NDRMSEs are not statistically significant at a 95% confidence interval for all three tropical ocean regions." How can you get the statement: "the research findings of this study demonstrate the potential of Aeolus observations on surface wind forecasts with the ECMWF model over the tropical ocean"?

**Response**: Thank you very much for pointing this out. We apologize for these conflicting statements.

The results for tropical oceans show some negative NCRMSEs within the T+144h, but unfortunately, the values are not statistically significant at a 95% confidence interval due to the limited number of data samples. We have modified the statements in the Sect.5 (Line: 762-769) and Sect.6 (Line: 908-910).

"According to the results of inter-comparison analyses for tropical oceans, the impact of Aeolus on sea surface wind forecast is nearly neutral overall. However, negative NCRMSE values are observed across all three ocean basins at the T+48 h forecast step. Despite not being statistically significant, this result is consistent with the verifications based on the model analysis at ECMWF (Rennie and Isaksen, 2022), but further demonstration is required with more in situ measurements."

"Notwithstanding the limited spatial coverage of the reference data, the research findings of this study provide information on the role of Aeolus data assimilation with the ECMWF model in near-surface wind forecasts over the tropical ocean and the high-latitude regions."

**Additional clarifications:**

In addition to addressing all concerns from the anonymous reviewers, we corrected a small error in data quality control for high-latitude regions and re-plotted all related figures. For TC analyses, we adjusted the method and re-processed the data without removing the outliers in order to make the results reflect the real forecast errors. Additionally, the normalized difference in root-mean-square error (NDRMSE) was changed to the normalized change in root-mean-square error (NCRMSE) to be consistent with the y-axis label of the plots for inter-comparison analysis.

**Reference:**

Abdalla, S., Isaksen, L., Janssen, P. A. E. M., and Wedi, N.: Effective spectral resolution of ECMWF atmospheric forecast models, ECMWF, Newsletter Number 137, 19–22, doi:10.21957/rue4o7ac, 2013.

Rennie, M. and Isaksen, L.: The NWP impact of Aeolus Level-2B winds at ECMWF, ECMWF, 227 pp., https://confluence.ecmwf.int/display/AEOL/L2B+team+technical+reports+and+relevant+papers?preview=/46596815/288355970/AED-TN-ECMWF-NWP-025--20220810_v5.0.pdf (last access: 20 October 2022), 2022.

Rennie, M. P., Isaksen, L., Weiler, F., Kloe, J., Kanitz, T., and Reitebuch, O.: The impact of Aeolus wind retrievals on ECMWF global weather forecasts, Q. J. R. Meteorol. Soc., 147, 3555–3586, https://doi.org/10.1002/qj.4142, 2021.

Ribal, A. and Young, I. R.: Global Calibration and Error Estimation of Altimeter, Scatterometer, and Radiometer Wind Speed Using Triple Collocation, Remote Sens., 12, 1997, https://doi.org/10.3390/rs12121997, 2020.

Vogelzang, J. and Stoffelen, A.: Triple collocation, Royal Netherlands Meteorological Institute, 22 pp., https://cdn.knmi.nl/system/data_center_publications/files/000/068/914/original/triplecollocation_nwpsaf_tr_kn_021_v1.0.pdf?1495621500 (last access: 27 January 2022), 2012.

---

## Author Response (AR2)

**Response to report #1 on amt-2022-311**

Anonymous Referee #2

**Suggestions for revision or reasons for rejection**

The revised manuscript is improved, but some concerns still remain.

We appreciate anonymous referee #2 for their effort in reviewing our revised manuscript and providing constructive comments.

Below are the responses to the comments and concerns from Referee #2.

(All concerns have been addressed point-by-point with responses highlighted in blue, and the corresponding modifications in orange have been incorporated into the revised manuscript. The line numbers mentioned in this response letter correspond to the revised manuscript of clean version.)

1. Since the impact of Aeolus winds on forecast are not statistically significant over the tropical oceans and the SHX, the result may be removed from the manuscript, just briefly mention the results in the manuscript.

**Response**: Thank you very much for this suggestion. We have removed some plots and texts for tropical ocean and Southern Hemisphere (SH) high-latitude regions. For example, for Figure 4 in the revised manuscript, we only keep the results for the tropical Pacific Ocean to show examples of the impact of Aeolus data quality on near-surface wind forecasts.

2. The Aeolus impact on the longer range forecast lead times in the NHX are statistically marginal significant. To understand and justify the results, and to prove your speculation: "For the high-latitude region in the Northern Hemisphere, the noticeable impact is found mainly from T+192 h onward, which is possibly owing to the downward propagation of Aeolus increments to the surface", the authors need to show further analysis demonstrating how the downward propagation is done in detail.

**Response**: Thank you very much for this comment.

To understand and justify the results, we made comparisons with existing studies. We find that our results are partly comparable with the verifications at ECMWF (Rennie and Isaksen, 2022). The main difference is that in our study, this evident positive impact exists at more forecast steps from T+192 h to T+240 h, which is partly due to the different reference data we are based on and the different spatial coverage they have.

Regarding the downward propagation, it would be worth doing a further analysis to demonstrate the downward propagation in the model. However, it is slightly beyond the scope of this paper, which mainly focuses on evaluating near-surface wind forecasts. Instead of doing further analysis or eliminating the sentence entirely, we tried to rephrase the sentences and added relevant references to support this speculation. Please see below:

"For the NH high-latitude region, Aeolus makes more positive impacts as the forecast extends. This result is partly comparable with the analysis-based verifications at ECMWF, with a noticeable positive impact obtained at the T+216 h forecast step (Rennie and Isaksen, 2022). The main difference is that in our study, this evident positive impact exists at more forecast

steps from T+192 h to T+240 h, which is in part due to the different reference data we are based on and the different spatial coverage they have. In addition, since there are a limited number of low-level Aeolus winds inland assimilated into the ECMWF model, we suspect that this positive impact is probably associated with the downward propagation of Aeolus increments to the surface as the changes in stratospheric initial conditions can affect tropospheric circulation on subsequent forecasts (Kodera et al., 1990; Christiansen, 2001; Charlton et al., 2004; Tripathi et al., 2015)."

(Lines 274-281)

3. The assumption of the independence of the errors of the two OSEs is questionable since the two OSEs are based on the same NWP system. You can show the actual correlations between the errors of the two OSEs to see if they are really small enough. Otherwise, the triple collocation results would be dropped.

**Response**: Thank you very much for this comment and suggestion.

We quantified the error correlations between the forecasts from two model runs and found the correlation coefficients are greater than 0.6 for most forecast steps. Thus, we have removed the results of triple collocation (TC) analyses. In addition, we added a paragraph in the Discussion section to explain the issues when implementing the TC analysis to assess two correlated data sets. Please see below:

"In terms of the evaluation method, apart from the conventional inter-comparison analysis like what we used in this study, triple collocation (TC) analysis is another beneficial method for environmental parameter evaluation when there are three independent data sets (Stoffelen, 1998; Vogelzang and Stoffelen, 2012). Different from the inter-comparison analysis that regards a reference data set free of errors, TC analysis assumes that each data set is linearly correlated with the truth. Following the equation derivation documented in Vogelzang and Stoffelen (2012), the primary output of TC is the error standard deviation (ESD) of each data set, which allows us to compare the quality of different data sets. We made an attempt to implement TC method to our cases (results are not shown). The results can generally reflect the impact of Aeolus on wind forecast, with the ESD from the forecast with Aeolus lower than the one without Aeolus implying the positive impact of Aeolus. But the ESD values are inaccurate since the errors of the two forecasts are not independent because they are from the same NWP model. Theoretically, without taking this dependence into account may lead to the ESDs of two forecasts under-estimated and the ESD of in situ measurements over-estimated since the error covariance of the two forecasts are greater than zero (Caires and Sterl, 2003). Therefore, to obtain accurate results when implementing the TC method to assess two correlated data sets, quantifying the non-zero covariance or making a further modification of the method is required."

(Lines 309-321)

**Reference:**

Caires, S. and Sterl, A.: Validation of ocean wind and wave data using triple collocation, J. Geophys. Res., 108, 3098, https://doi.org/10.1029/2002JC001491, 2003.

Charlton, A. J., Oneill, A., Lahoz, W. A., and Massacand, A. C.: Sensitivity of tropospheric forecasts to stratospheric initial conditions, Q. J. R. Meteorol. Soc., 130, 1771–1792, https://doi.org/10.1256/qj.03.167, 2004.

Christiansen, B.: Downward propagation of zonal mean zonal wind anomalies from the stratosphere to the troposphere: Model and reanalysis, J. Geophys. Res., 106, 27307–27322, https://doi.org/10.1029/2000JD000214, 2001.

Kodera, K., Yamazaki, K., Chiba, M., and Shibata, K.: Downward propagation of upper stratospheric mean zonal wind perturbation to the troposphere, Geophys. Res. Lett., 17, 1263–1266, https://doi.org/10.1029/GL017i009p01263, 1990.

Rennie, M. and Isaksen, L.: The NWP impact of Aeolus Level-2B winds at ECMWF, ECMWF, 227 pp., https://confluence.ecmwf.int/display/AEOL/L2B+team+technical+reports+and+relevant+papers?preview=/46596815/288355970/AED-TN-ECMWF-NWP-025--20220810_v5.0.pdf (last access: 20 October 2022), 2022.

Stoffelen, A.: Toward the true near-surface wind speed: Error modeling and calibration using triple collocation, J. Geophys. Res., 103, 7755–7766, https://doi.org/10.1029/97JC03180, 1998.

Tripathi, O. P., Baldwin, M., Charlton-Perez, A., Charron, M., Eckermann, S. D., Gerber, E., Harrison, R. G., Jackson, D. R., Kim, B., Kuroda, Y., Lang, A., Mahmood, S., Mizuta, R., Roff, G., Sigmond, M., and Son, S.: The predictability of the extratropical stratosphere on monthly time-scales and its impact on the skill of tropospheric forecasts, Q.J.R. Meteorol. Soc., 141, 987 – 1003, https://doi.org/10.1002/qj.2432, 2015.

Vogelzang, J. and Stoffelen, A.: Triple collocation, Royal Netherlands Meteorological Institute, 22 pp., https://cdn.knmi.nl/system/data_center_publications/files/000/068/914/original/triplecollocation_nwpsaf_tr_kn_021_v1.0.pdf?1495621500 (last access: 27 January 2022), 2012.

**Response to report #2 on amt-2022-311**

Anonymous Referee #1

**Suggestions for revision or reasons for rejection**

The authors have addressed my comments and improved the manuscript significantly. They also removed all ambigous interpretations of the results by clearly stating where the assimilation improvements are not significant.

I very much support the inclusion of the new Figure 1, but would suggest to change the longitudinal sampling from 5° to a multiple of the Aeolus orbit distance (~3.2°) to avoid the strange looking checkerboard pattern. Once this change is applied, I think the article is ready for publishing.

We are grateful for the positive feedback and suggestion from anonymous Referee #1 on our revised manuscript.

The map in Figure 1 has been re-generated with a grid size of 3.2°x3.2°.

[Figure]

Figure 1. The averaged number of L2B Mie-cloudy winds at pressure > 850 hPa assimilated into the model